# CFP1 governs uterine epigenetic landscapes to intervene in progesterone responses for uterine physiology and suppression of endometriosis

Seung Chel Yang [1,10], Mira Park [1,10], Kwon-Ho Hong [2], Hyeonwoo La[2], Chanhyeok Park[2], Peike Wang[3], Gaizhen Li[3], Qionghua Chen[3], Youngsok Choi [2], Francesco J. DeMayo[4], John P. Lydon[5], David G. Skalnik[6], Hyunjung J. Lim [7], Seok-Ho Hong [8,9], So Hee Park [1], Yeon Sun Kim[1], Hye-Ryun Kim[1] & Haengseok Song [1] ✉

Progesterone ($P_4$) is required for the preparation of the endometrium for a successful pregnancy. $P_4$ resistance is a leading cause of the pathogenesis of endometrial disorders like endometriosis, often leading to infertility; however, the underlying epigenetic cause remains unclear. Here we demonstrate that CFP1, a regulator of H3K4me3, is required for maintaining epigenetic landscapes of $P_4$-progesterone receptor (PGR) signaling networks in the mouse uterus. *Cfp1*^f/f^;*Pgr*-Cre (*Cfp1*^d/d^) mice showed impaired $P_4$ responses, leading to complete failure of embryo implantation. mRNA and chromatin immunoprecipitation sequencing analyses showed that CFP1 regulates uterine mRNA profiles not only in H3K4me3-dependent but also in H3K4me3-independent manners. CFP1 directly regulates important $P_4$ response genes, including *Gata2*, *Sox17*, and *Ihh*, which activate smoothened signaling pathway in the uterus. In a mouse model of endometriosis, *Cfp1*^d/d^ ectopic lesions showed $P_4$ resistance, which was rescued by a smoothened agonist. In human endometriosis, CFP1 was significantly downregulated, and expression levels between CFP1 and these $P_4$ targets are positively related regardless of PGR levels. In brief, our study provides that CFP1 intervenes in the $P_4$-epigenome-transcriptome networks for uterine receptivity for embryo implantation and the pathogenesis of endometriosis.

CXXC finger protein 1 (CFP1) is an important player in the epigenetic regulation of genes by inducing trimethylation at histone H3 lysine 4 (H3K4me3) with SETD1, a histone methyltransferase[1]. H3K4me3 is mainly found in active promoters and turns chromatin into transcriptionally active euchromatin in the transcription start site (TSS) and CpG island (CGI)[2]. When CFP1 binds to unmethylated CpG, it recruits SETD1A/B to trigger H3K4me3 in the promoters of target genes to increase gene expression[1,3]. The deletion of CFP1, SETD1A, or SETD1B in mice caused embryonic lethality during or after gastrulation, suggesting the roles of SETD1-CFP1 complexes in early mammalian development[4,5]. Furthermore, the conditional deletion of CFP1 highlighted that CFP1-associated H3K4me3 has fundamental roles in various biological processes. CFP1-deficient embryonic stem cells failed to differentiate in vitro because of aberrant H3K4me3 at

---

non-methylated CGI promoters, leading to transcriptional disturbance[2,4,6]. CFP1 plays an important role in intrathymic T-cell development and differentiation program of TH17 cells[7,8]. Recent studies have also demonstrated that CFP1 is required for epigenetic modification in non-replicative cells, such as germ cells, in mice[9–11].

The endometrium is a highly dynamic organ, and its function and cyclicity are mainly regulated by ovarian progesterone ($P_4$) and estrogen ($E_2$). Imbalances between $P_4$ and $E_2$ may cause various gynecological disorders such as endometriosis, repeated implantation failure, and endometrial cancer[12]. $P_4$ and $E_2$ activate their nuclear receptors, progesterone receptors (PGRs), and estrogen receptors (ESRs) to control the expression of local factors for uterine functions. PGR-dependent signaling networks for epithelium–stroma interaction are necessary to promote uterine receptivity for embryo implantation and decidualization in the uterus[13]. The fine-tuning of epigenetic modulation is required for phase-specific transcriptional networks during the reproductive cycle, embryo implantation, and subsequent pregnancy[14–16]. Dynamic changes in histone modifications occur during decidualization[17,18]. Silencing and overexpression of EZH2, a histone methyltransferase for H3K27me3, a repressive histone mark, disturb the expression of decidualization markers, such as IGFBP1 and PRL[17]. For successful parturition, biphasic modes of H3K27me3 dynamics in decidual stromal cells dictate the regulated gene silencing in the uterine adaptation to pregnancy[19]. The acetylation and methylation patterns of H3 and H4 in women with endometriotic lesions are distinct from those of disease-free women[20]. However, the physiological significance of epigenetic regulatory machineries, such as histone modification during early pregnancy and the pathogenesis of uterine disorders, remains largely unexplored.

Endometriosis is a disorder in which endometrial cells grow abnormally outside the uterus. The surgical removal of the ectopic endometrial lesion with hormonal suppression is the current standard of care; however, these therapies have a high incidence of relapse and various side effects[21,22]. Endometriosis affects 10%–15% of women of reproductive age and is one of the leading causes of female infertility[21–24]. Leading causes of endometriosis include increased $E_2$ response, $P_4$ resistance (decreased $P_4$ response), and/or abnormal epigenetic regulation[25–28]. Although decreased $P_4$ response caused by reduced PGR expression is considered a main cause of endometriosis[26,29], the underlying cause of the decreased $P_4$ response, even with normal $P_4$ secretion and PGR expression, has not yet been elucidated.

Altered DNA methylation and histone modification on the genes for balanced hormone responses were mainly proposed to affect the endometrial function and the development of endometriosis[9,12]. The loss of histone deacetylase 3 (HDAC3) is a result of failures in embryo implantation and decidualization and caused fibrosis in the endometrium, one of the symptoms of human endometriosis with reduced HDAC3 expression[25]. However, epigenetic causes of how uterine cells grow outside the uterus in patients with endometriosis have not been clearly identified. Using a combination of genetic and pharmacologic tools and various analyses, we demonstrated that CFP1-dependent epigenetic regulation is necessary to maintain uterine transcriptional landscapes for $P_4$ response for a successful pregnancy and prevent endometriosis with $P_4$ resistance. Thus, this study provides fundamental insight into understanding the complex interplay between the $P_4$-PGR signaling pathway and uterine epigenome–transcriptome under physiologic and pathophysiologic conditions, such as endometriosis.

## Results
### Loss of *Cfp1* leads to infertility with multiple failures in oviductal embryo transport, $P_4$ uterine responses, embryo implantation, and decidualization in mice

CFP1 is expressed in many cells of female reproductive organs, such as the ovary, oviduct, and uterus (Supplementary Fig. 1). During early pregnancy, uterine CFP1 expression gradually increased from day 1 of pregnancy (Day 1) to Day 3. It peaked on Day 4, with the highest expression in the luminal epithelium. However, its expression is not directly affected by the regulation of $E_2$ and/or $P_4$ in the mouse uterus (Supplementary Fig. 1). While *Cfp1* is deleted in most cell types in the female reproductive tract of adult *Cfp1*[f/f];*Pgr*[cre/+] (*Cfp1*[d/d]) mice (Supplementary Fig. 2), they showed normal architectures in the gross morphology and histology of the reproductive tract with regular estrous cycle (Supplementary Fig. 3a–c). Furthermore, serum levels of $E_2$ and $P_4$ on Day 4 in *Cfp1*[d/d] mice were comparable to those of *Cfp1*[f/f] mice. We also found that *Cfp1*[d/d] mice ovulate similar numbers of oocytes that can fertilize normally, and the fertilized embryos develop to the blastocyst stage without any aberrations in vitro (Supplementary Fig. 3d–f). However, implantation sites (IS), blue bands along the uterus, were not observed in *Cfp1*[d/d] female mice on Day 5 (Fig. 1a). In addition, *Cfp1*[d/d] uteri did not show decidual responses to artificial stimuli, such as oil (Fig. 1b, c). Accordingly, *Cfp1*[d/d] female mice did not produce any pups (Fig. 1d). Interestingly, all blastocysts were found in the oviduct but not in the uterus of *Cfp1*[d/d] mice on Day 4, whereas they were found in the uterus of *Cfp1*[f/f] mice as expected (Fig. 1e–g).

To examine embryo implantation in the uterus of *Cfp1*[d/d] mice with defective oviductal embryo transport, wildtype blastocysts were transferred to the uteri of *Cfp1*[f/f] and *Cfp1*[d/d] mice on day 4 of pseudopregnancy. Distinct IS were observed in *Cfp1*[f/f] but not in *Cfp1*[d/d] recipients, and unimplanted blastocysts were retrieved from *Cfp1*[d/d] recipients 24 h after embryo transfer (Fig. 1h, i and Table 1), indicating that CFP1-deficient uterine environments do not support embryo implantation. In this aspect, cell proliferation profiles are aberrant in the uterus of *Cfp1*[d/d] mice on Day 4 (Fig. 1j, k), whereas the levels of $E_2$ and $P_4$ (Supplementary Fig. 3d) and their receptors (Supplementary Fig. 4) are comparable between *Cfp1*[f/f] and *Cfp1*[d/d] mice. Nevertheless, the uterine epithelium persistently proliferates, and the stroma showed less proliferation potential in the uterus of *Cfp1*[d/d] mice, suggesting that CFP1, as an epigenetic regulator, is required for proper hormone responses in the uterus for a successful pregnancy.

### CFP1 regulates uterine mRNA landscapes in both H3K4me3-dependent and H3K4me3-independent manners
To understand the CFP1-dependent epigenetic regulation on transcriptional landscapes in the uterus, we performed chromatin immunoprecipitation sequencing (ChIP-seq) and mRNA sequencing (mRNA-seq) with the uteri of *Cfp1*[f/f] and/or *Cfp1*[d/d] mice on Day 4 (Fig. 2). ChIP-seq with antibodies for CFP1 and H3K4me3 provided evidence that CFP1-binding sites are highly enriched in TSS and CGI, and H3K4me3 in TSS and CGI was generally reduced in the uterus of *Cfp1*[d/d] mouse (Fig. 2a). De novo motif analysis for CFP1 ChIP-seq data showed that CFP1 recognizes the CCGG motifs, including CGG and its reverse complement CCG (Fig. 2b), which are consistent with the CFP1-binding motifs found in humans[30–32]. CFP1 binding was noticeably enriched in extended gene bodies, including promoters (9%), exons (4%), and introns (38%) in mouse uterus, considering that the mouse genome consists of promoters (2%), exons (2%), introns (20%), and intergenic factors (76%) (Fig. 2c). When the mRNA-seq and H3K4me3 ChIP-seq data in ±2 kilo base pairs (Kbp) of TSS were analyzed together, 40.3% of differentially expressed genes (DEGs) had a reduction in both gene expression and H3K4me3 levels in *Cfp1*[d/d] mice (Fig. 2d) as expected from the known actions of CFP1 on H3K4me3. Generally, the lower gene expression levels are, the lower the H3K4me3 levels in the +2Kbp TSS region are (Supplementary Fig. 5). However, a significant portion of DEGs (18.2%) was made up of genes with decreased expression levels and increased H3K4me3 levels in *Cfp1*[d/d] mouse uterus on Day 4, suggesting that CFP1 could promote gene expression in the uterus in H3K4me3-independent manner.

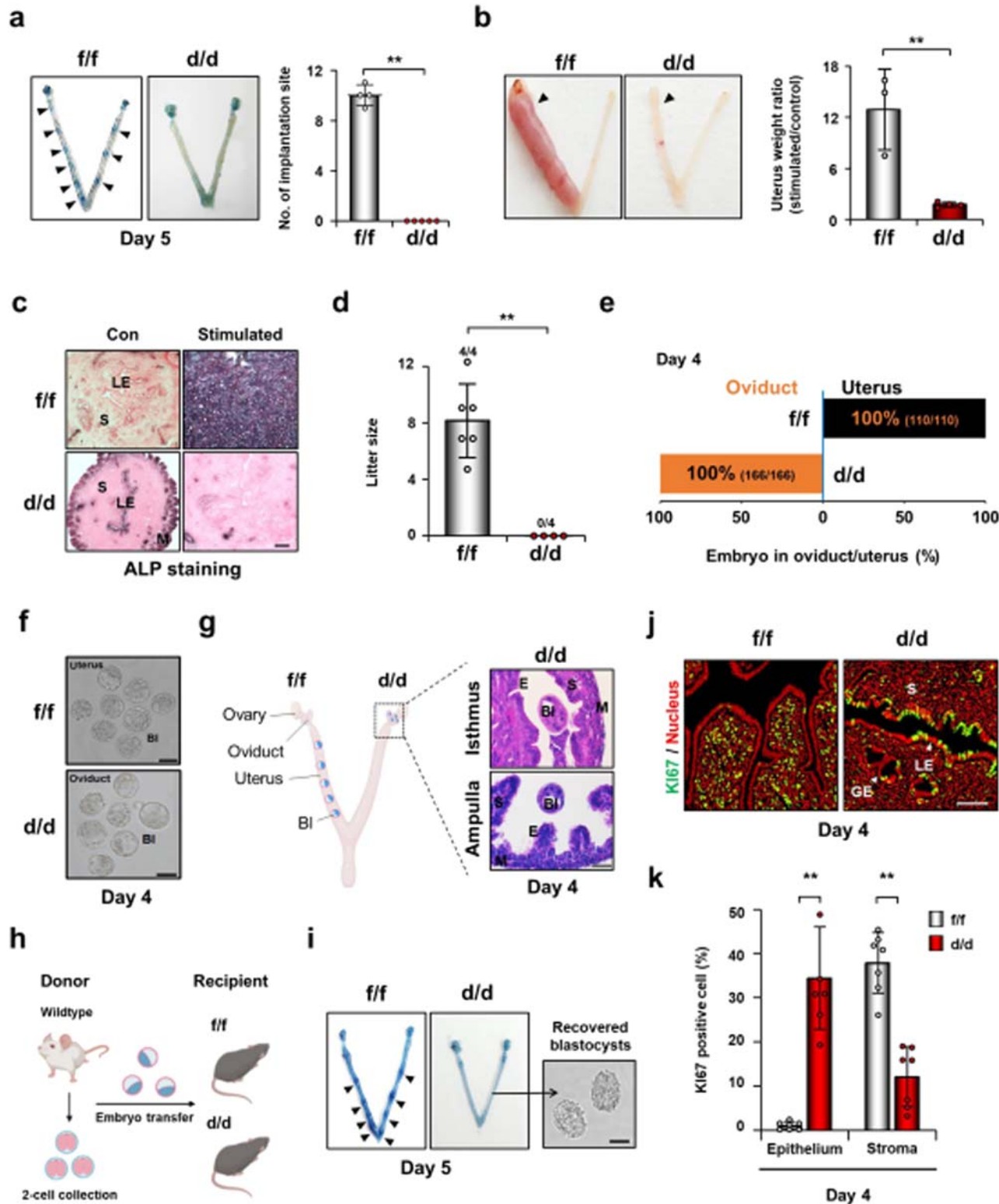

Although CFP1 binds to unmethylated CpG and induces H3K4me3 to increase gene expression[1,3], the number of significantly down-regulated (3329) and upregulated (2829) genes in $Cfp1^{d/d}$ mice was comparable (54% vs. 46%) in DEGs from mRNA-seq (Fig. 2e and Supplementary Fig. 6a). However, gene ontology (GO) analyses showed that most GO terms (126/132, 95.5%) with false discovery rate (FDR) of <0.25 were reduced in the uteri of $Cfp1^{d/d}$ mice on Day 4 (Fig. 2f and Supplementary Fig. 6b). Gene set enrichment analyses (GSEA) showed that various gene sets associated with $P_4$ response, hedgehog

signaling, cancers, ion channel activities, and stromal cell stimulation were significantly downregulated in $Cfp1^{d/d}$ uterus (Fig. 2g). Interest-ingly, the Indian hedgehog (IHH)-dependent smoothened signaling pathway (SSP) is a well-known $P_4$ downstream pathway that inhibits $E_2$-dependent epithelial proliferation and further stimulates stromal proliferation in the uterus[33]. A heatmap, reverse-transcriptase poly-merase chain reaction (RT-PCR), and real-time RT-PCR analyses for the SSP gene set validated that IHH-dependent SSP is significantly down-regulated in $Cfp1^{d/d}$ mouse uterus on Day 4 (Fig. 2h, i).

**Fig. 1 | Cfp1<sup>d/d</sup> mice suffer from aberrant epithelial cell proliferation and complete failure of embryo implantation and decidualization. a** Representative photographs of uteri with IS (black arrowheads) in *Cfp1*<sup>f/f</sup> and *Cfp1*<sup>d/d</sup> mice on Day 5. *n* = 4 to 5 biologically independent samples per genotype. Data are presented as mean values with SD. Statistical analyses were performed using the unpaired Student's *t*-tests. **p < 0.01. **b** Artificial decidualization responses in hormone primed *Cfp1*<sup>f/f</sup> and *Cfp1*<sup>d/d</sup> OVX mice. The decidual response was determined by the uterine weight of the oil-injected (black arrowheads)/non-injected uterine horn. *n* = 3 to 4 biologically independent samples per genotype. Data are presented as mean values with SD. Statistical analyses were performed using the unpaired Student's *t*-tests. **p < 0.01. **c** Microscopic images of alkaline phosphatase staining of artificially decidualized *Cfp1*<sup>f/f</sup> and *Cfp1*<sup>d/d</sup> uteri. Scale bar, 100 μm. **d** Litter size of *Cfp1*<sup>f/f</sup> and *Cfp1*<sup>d/d</sup> female mice that were mated with fertile male mice for 8–10 weeks. The numbers above the bars indicate the number of mice with litter/total number of mice examined in each group. Data are presented as mean values with SD. Statistical analyses were performed using the unpaired Student's *t*-tests. **p < 0.01.

**e–h** Impairment of the embryo transport from the oviduct to the uterus in *Cfp1*<sup>d/d</sup> female mice. Percentage graph (**e**) and microscopic images (**g**) of embryos recovered from uteri and/or oviducts in *Cfp1*<sup>f/f</sup> and *Cfp1*<sup>d/d</sup> mice on Day 4. Scale bar, 50 μm in (**f**). **g** Schematic image of *Cfp1*<sup>d/d</sup> oviduct in the morning of Day 4. Representative histological images of *Cfp1*<sup>d/d</sup> oviduct (ampulla and isthmus) where blastocysts were found even in the morning of Day 4. Scale bar, 50 μm. Figure was created with BioRender.com. **h** Experimental scheme of embryo transfer. **i** Representative photographs of uteri with IS (black arrowheads) in pseudopregnant *Cfp1*<sup>f/f</sup> and *Cfp1*<sup>d/d</sup> recipients after transferring wildtype blastocysts. Scale bars, 50 μm. Figure was created with BioRender.com. **j, k** Immunofluorescent staining of KI67 to examine uterine cell proliferation in *Cfp1*<sup>f/f</sup> and *Cfp1*<sup>d/d</sup> mice on Day 4. Scale bar, 50 μm. Bl blastocyst, S stroma, M muscle cells, E epithelium, GE glandular epithelium, LE luminal epithelium. *n* = 6 to 7 biologically independent samples per genotype. Data are presented as mean values with SD. Statistical analyses were performed using the unpaired Student's *t*-tests. **p < 0.01.

**Table 1 | Embryo implantation in pseudopregnant *Cfp1*<sup>f/f</sup> and *Cfp1*<sup>d/d</sup> recipients after blastocyst transfer**

| Genotype of embryo | Genotypes of recipients | No. of recipients (No. of transferred embryos) | No. of mice with IS (%) | No. of IS (%) | No. of blastocysts recovered (%) |
|---|---|---|---|---|---|
| Wildtype | *Cfp1*<sup>f/f</sup> | 5 (76) | 5 (100.0) | 7.2 ± 1.9 (47.4) | N.A. |
| | *Cfp1*<sup>d/d</sup> | 8 (126) | 0 (0.0) | 0 (0.0) | 0.4 ± 0.7 (2.4) |

*No number, IS implantation site, N.A. not applicable.*

## CFP1 epigenetically regulates $P_4$ responses through the IHH-dependent SSP in mouse uterus

To further investigate the epigenetic actions of CFP1 in the uterus, we tried to identify CFP1 direct target genes by analyzing all sequencing data together. The CFP1 direct target gene candidates should have CFP1-binding site(s) and be downregulated in *Cfp1*<sup>d/d</sup> uteri. We found that the putative direct target genes of CFP1 are regulated not only in H3K4me3-dependent (673 genes) but also in H3K4me3-independent (423 genes) manner (Fig. 3a and Supplementary Data 1–2). Interestingly, the upstream and downstream genes of IHH-dependent SSP were included in the list; *Ihh*, *Gli3*, and *Gata2* as H3K4me3-dependent and *Ptch1*, *Sox17*, and *Nr2f2* as H3K4me3-independent target genes (Fig. 3b and Supplementary Data 2). The expression of these genes was reduced with statistical significance in *Cfp1*<sup>d/d</sup> mice (Figs. 2i and 3c). The visualization of sequencing data for *Gata2*, *Ihh*, and *Sox17* using Integrative Genomics Viewer demonstrated that they all have CFP1 binding sites, and their expression levels were significantly reduced in *Cfp1*<sup>d/d</sup> uteri. However, H3K4me3 levels were decreased in extended gene bodies of *Gata2* and *Ihh*, but not in *Sox17* (Fig. 3d). Real-time ChIP PCR reinforced that SETD1 and H3K4me3 were enriched in the promoter regions of *Gata2* and *Ihh* but not in *Sox17*, whereas CFP1 is enriched in all of them in *Cfp1*<sup>f/f</sup> uterus (Fig. 3e and Supplementary Table 1). In summary, CFP1 works with SETD1 to increase H3K4me3 in *Gata2* and *Ihh* promoters but not for *Sox17* promoter for their expression in the uterus.

$P_4$-PGR induces stromal cell proliferation and inhibits epithelial cell proliferation via the activation of SSP downstream of *Gata2*, *Sox17*, and *Ihh* in mouse uterus on Day 4 (Fig. 3b). However, $P_4$ could not inhibit $E_2$-dependent epithelial proliferation and facilitate stromal cell proliferation in the uteri of ovariectomized (OVX) *Cfp1*<sup>d/d</sup> mice (Fig. 3f, g) in line with the results on Day 4 (Fig. 1k). Since defective SSP could cause aberrant cell proliferation in *Cfp1*<sup>d/d</sup> uteri, we tried to rescue this phenotype with SAG, a smoothened agonist, in these mice. An intrauterine delivery of SAG successfully rescued aberrant epithelial proliferation in the uteri of OVX *Cfp1*<sup>d/d</sup> mice treated with $E_2$ and $P_4$ (Fig. 3f and 3g). Furthermore, SAG significantly restored the decreased expression levels of genes in SSP and its downstream genes, such as *Gli1, Gli2, Nr2f2*, and *Hand2*, in *Cfp1*<sup>d/d</sup> uteri to the levels in *Cfp1*<sup>f/f</sup> uteri (Fig. 3h). When our mRNA-seq datasets were compared with other $P_4$-related transcriptomic data (GSE118264, GSE40661, and GSE178541)[13,34,35],

comparative analyses showed that significant numbers of DEGs in *Cfp1*<sup>d/d</sup> uterus overlap with DEGs in other $P_4$-related datasets (Supplementary Fig. 7). Collectively, these results suggest that CFP1 loss disturbs the epigenetic maintenance of $P_4$-PGR signaling pathways in the uterus.

## CFP1 is required for $P_4$ function to inhibit the growth of ectopic endometriotic lesions in mice

Endometriosis can occur possibly via a decreased $P_4$ response, and some patients with endometriosis show $P_4$ resistance even with normal *PGR* expression[36–39]. To further evaluate CFP1 function for uterine $P_4$ responses, we established a mouse model of endometriosis with some modifications from previous reports[25,26]. When $P_4$ was given with $E_2$, the size of ectopic lesions was significantly smaller than with $E_2$ alone, although the number of ectopic lesions was not different (Supplementary Fig. 8a, b). Ectopic lesions were mainly observed on organs with highly developed blood vessels, such as the small intestine, kidney, uterus, and fat pad (Supplementary Fig. 8c). When small pieces of *Cfp1*<sup>f/f</sup> and *Cfp1*<sup>d/d</sup> uterus as endometriotic lesions were transplanted to wildtype recipients (Fig. 4a), $P_4$ effectively suppressed the $E_2$-induced growth of *Cfp1*<sup>f/f</sup> but not *Cfp1*<sup>d/d</sup> ectopic lesions even if the number of ectopic lesions was not affected by *Cfp1* genotypes (Fig. 4b–e). The mRNA expression of *Gata2*, *Sox17*, and *Ihh* was also significantly downregulated in *Cfp1*<sup>d/d</sup> ectopic uterine lesions (Fig. 4f). When SAG was administered to rescue $P_4$ resistance and/or insensitivity in *Cfp1*<sup>d/d</sup> ectopic uterine lesions, $P_4$ suppressed the size of *Cfp1*<sup>d/d</sup> ectopic lesions in the endometriosis model (Fig. 4b–e), suggesting that CFP1 is required for proper $P_4$ responses to suppress ectopic growth of uterine tissues in mice.

## Downregulation of the epigenetic factor *CFP1* may be associated with endometriosis in humans

To further evaluate the potential actions of *CFP1* on the pathogenesis of endometriosis in humans, we analyzed datasets (GSE51981) of the endometria of patients with endometriosis (endometriosis group, *n* = 77) and healthy women (control group, *n* = 71) from a previous study[40]. While no correlation exists between the expression levels of *CFP1* and *PGR* mRNAs, *GATA2*, *SOX17*, and *IHH* mRNA expression levels were positively correlated with that of *CFP1* (Fig. 5a) and *PGR* mRNA (Supplementary Fig. 9a) regardless of endometrial pathologic

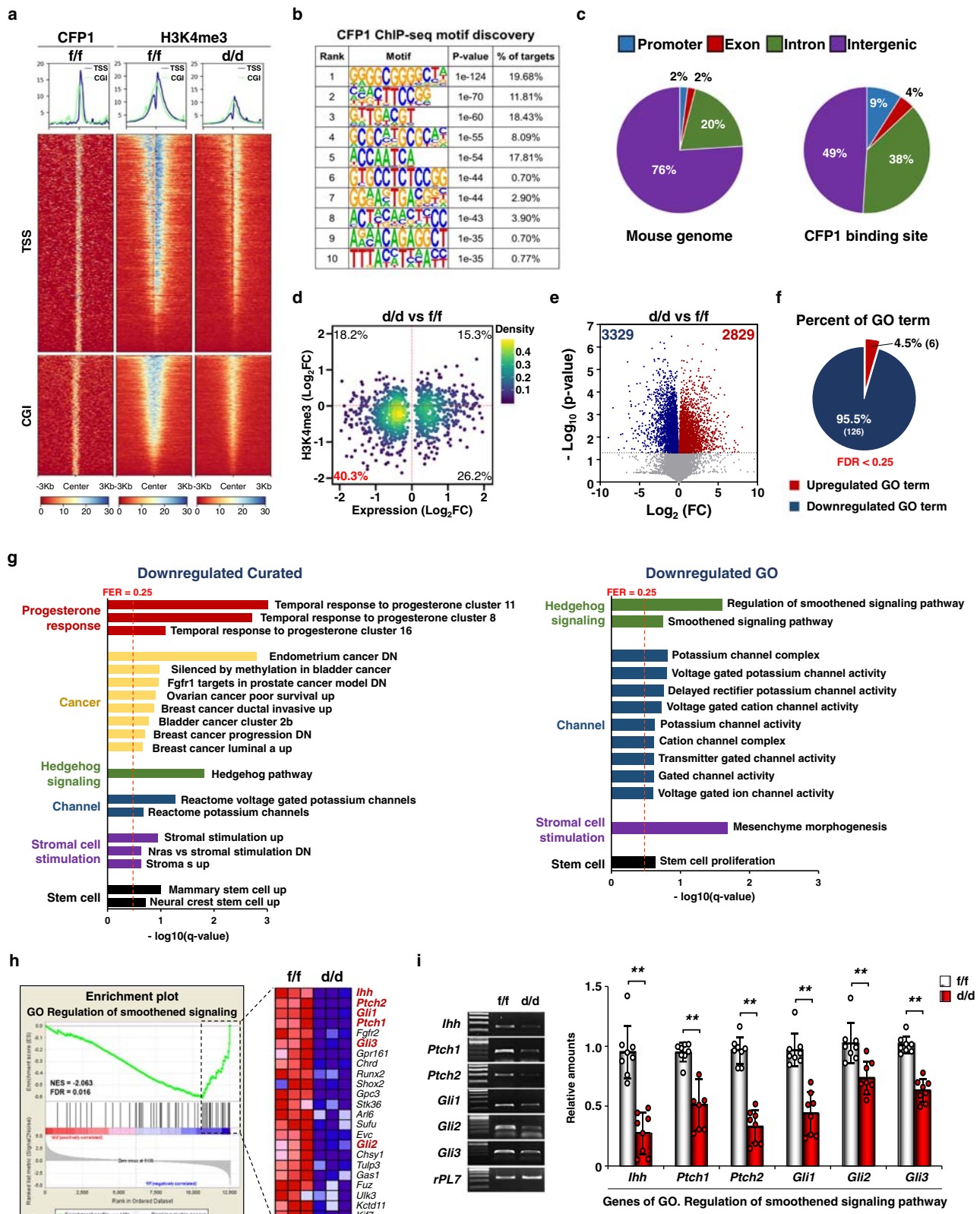

conditions. We further analyzed a subset of GSE51981, i.e., endometria of women with mild and severe endometriosis in the mid-secretory phase where P4 is dominant (Fig. 5b). In patients with mild endometriosis, the expression profiles of *PGR*, *CFP1*, and P4 target genes are not different from those of the control group. However, there was a statistically significant reduction in their expression levels in patients with severe endometriosis (Supplementary Fig. 9b). Among patients with

severe endometriosis, some had a comparable level of *PGR* but a low level of *CFP1* (circled red, Fig. 5b). In these patients, the expression patterns of *GATA2*, *SOX17*, and *IHH* mRNAs were significantly decreased. It suggests that aberrantly reduced expression of *CFP1* may be associated with endometriosis via abnormal P4 response in patients with normal *PGR* expression. We also performed immunohistochemistry and real-time RT-PCR for these genes in endometrial samples

**Fig. 2 | CFP1 epigenetically regulates uterine transcriptome via H3K4me3 on Day 4. a** Tag density of CFP1 binding and H3K4me3 peaks were calculated on ±3 Kbp window centered on TTS and CGI regions of all RefSeq (mm10) genes in *Cfp1*[f/f] and/or *Cfp1*[d/d] mouse uteri on Day 4 for heatmap and graph data. **b** CFP1-binding sequence logo of the top 10 motifs identified using de novo motif discovery. **c** Distribution of the genetic features across the mouse genome and CFP1-binding peaks in *Cfp1*[f/f] mouse uterus on Day 4. **d** Correlation between H3K4me3 promoter (TSS ± 2 Kbp) enrichment conditions and gene expression in the mouse uterus on Day 4 (*Cfp1*[d/d] versus *Cfp1*[f/f]). Each dot represents a differentially expressed gene with statistical significance (*p* < 0.05, normalized data average, log$_2$ > 3). **e** Volcano plot to compare expression profiles from the *Cfp1*[f/f] vs. *Cfp1*[d/d] in the uterus on Day

4. **f** Pie chart summarizing upregulated or GO term in *Cfp1*[f/f] versus *Cfp1*[d/d] mouse uterus on Day 4. **g** GSEA to identify downregulated GO term and curated gene sets in *Cfp1*[f/f] versus *Cfp1*[d/d] mouse uterus on Day 4. Gene sets with an FDR *q*-value of <0.25 (red dotted line) were considered significant. **h** GSEA enrichment plot and heatmap of the "GO Regulation of smoothened signaling" gene set from RNA-seq data of *Cfp1*[f/f] and *Cfp1*[d/d] mouse uterus on Day 4. The color spectrum from blue to red indicates low to high expression. **i** RT-PCR and Real-time RT-PCR analyses for IHH-dependent SSP genes (red colored in **h**). *n* = 8 biologically independent samples per genotype. Data are presented as mean values with SD. Statistical analyses were performed using the unpaired Student's *t*-tests. **p* < 0.01.

from patients with severe endometriosis and disease-free control women in the secretory phase. The eutopic and ectopic endometria from patients with endometriosis showed decreased expression levels of *CFP1*, *GATA2*, *SOX17*, and *IHH* mRNAs compared with the endometrium from the control group (Fig. 5c, d), suggesting that reduced CFP1 expression may disturb epigenetic landscapes that mediate P$_4$-PGR signaling pathways in human endometrium, often leading to endometriosis.

## Discussion

Many gynecological diseases are caused by abnormally reduced P$_4$ response; however, the underlying epigenetic aberration for the imbalanced steroid hormone responses has not been clearly elucidated. We demonstrate that CFP1-associated epigenetic regulation is required for maintaining appropriate P$_4$ responses for embryo implantation and decidualization in the uterus and inhibiting the ectopic growth of endometrial lesions outside the uterus. *Cfp1*[d/d] mice exhibit a wide spectrum of infertility, including defective oviductal embryo transport, abnormal uterine cell proliferation, and complete failure of embryo implantation and decidualization (Fig. 1). These phenotypes suggest that epigenetic regulation through H3K4me3 is involved in the sequential events of female reproduction. A recent study supported those spatiotemporal dynamics of H3K4me3 in the uterine genome are needed to be well controlled for early pregnancy. In mice without MENIN (*Men1*[d/d] mice), a member of the H3K4 methyltransferase complex, defective decidualization compromised fertility, whereas embryo implantation normally occurs[41]. Impaired decidualization in *Men1*[d/d] mice was caused by reduced *Bmp2* expression by abnormally increased FGF2 signaling[41]. In *Cfp1*[d/d] uterus on Day 4, *Fgf2* expression aberrantly increased because of a decrease in *Ihh*-dependent SSP that antagonizes FGF2 signaling, although the *Bmp2* expression did not decrease (Fig. 3). While MLL1/2 complexes that contain MENIN catalyze H3K4 methylation in a gene- and cell-specific manner, SETD1–CFP1 complexes are the leading H3K4 methyltransferases among the six histone methyltransferases, SETD1A/B, MLL1/2, and MLL3/4[9,41]. This is consistent with the fact that *Cfp1*[d/d] mice show a wider spectrum of infertile phenotypes than *Men1*[d/d] mice, suggesting that CFP1-dependent epigenetic regulation may work on the wider area of the genome. For example, we found an impairment of the oviductal embryo transport in *Cfp1*[d/d] mice, which was not observed in *Men1*[d/d] mice. Embryo transport in the oviduct could be interrupted by abnormalities in cilia movement, fluid secretion, and smooth muscle contraction[42,43]. Since *Cfp1* was mainly deleted in non-ciliated epithelial cells of the isthmus, but not in ciliated ones, which are positive for acetylated tubulin in the ampulla of *Cfp1*[d/d] oviduct by *Pgr*-Cre (Supplementary Fig. 2e, f), embryo retainment in the oviduct could result from abnormalities in fluid secretion and smooth muscle contraction[44], but not from cilia movement. While the epigenetic changes for fluid secretion are largely unknown, histone marks to promote transcription, such as H3K4me3 and H3K27 acetylation (H3K27ac), are enriched at promoters of genes driving muscle contraction on the advance of labor onset in the myometrium[45].

CFP1-dependent epigenetic regulation has been investigated in various biological events. The conditional deletion of *Cfp1* in mouse hematopoietic cells resulted in severe defects during hematopoiesis with complete loss of lineage-committed progenitors and mature cells[46]. CFP1 is also required for thymocyte survival, the balanced differentiation between Th17 and Treg cells[7,8], and the phagocytic and bactericidal activity of macrophages[47]. Since *Pgr* is expressed in various immune cells, including macrophages[48], dendritic cells[49], and T cells[50], *Cfp1* was supposed to be deleted in immune cells as well as uterine cells in *Cfp1*[d/d] mice. Immune cells play important roles during pregnancy[51]. However, immune-related gene sets were not significantly altered in *Cfp1*[d/d] mouse uterus on Day 4 (Fig. 2g). Furthermore, normal fertility was observed in mice (*Cfp1*[f/f];*LysM*[cre/+]) in which *Cfp1* is deleted in the myeloid lineage cells, such as monocytes and macrophages, using *LysM*-Cre (Supplementary Table 2), suggesting that phenotypes observed in *Cfp1*[d/d] mice are not directly associated with CFP1 functions in immune cells. Recently, the function of CFP1–SETD1 complexes for epigenetic reprogramming during germ cell development has been investigated. CFP1-mediated H3K4me3 is required for maintaining chromatin accessibility for transcriptional activities during oocyte development, and oocyte-specific deletion of *Cfp1* caused reduced H3K4 methylation levels and globally downregulated transcription activities, in turn, leading to multiple defects in the meiotic division and maternal–to–zygotic transition[10]. Furthermore, CFP1 participates in regulating the expression of paracrine factors for communication between the oocyte and surrounding granulosa cells for follicle growth and ovulation[52]. The conditional deletion of *Setd1b* in the oocyte also caused the dysregulation of transcription factors for oogenesis, including *Obox* transcription factors, leading to oocyte maturation defects and infertility[53]. CFP1 deletion before the onset of meiosis with *Stra8*-Cre in male mice caused complete infertility with the spermatogenic arrest at the MII stage[11], suggesting that CFP1-mediated H3K4me3 plays a role in meiosis and cell fate decision.

P$_4$ antagonizes E$_2$ action on epithelial proliferation and promotes stromal cell proliferation to prepare uterine receptivity for embryo implantation and decidualization[13,54]. P$_4$ increased the expression of *Gata2*, *Sox17*, and *Ihh* in epithelial cells to trigger SSP in the stroma on Day 4 in mice[13,33,34], all of which were downregulated in *Cfp1*[d/d] mice (Figs. 2 and 3). As a result, uterine epithelial cells aberrantly proliferated and failed to prepare embryo implantation on Day 4 in *Cfp1*[d/d] mice (Fig. 1). The expression of gene sets related to P$_4$ response, IHH-dependent SSP, and stromal cell stimulation was significantly reduced in *Cfp1*[d/d] mice (Fig. 2g). Furthermore, SAG successfully rescued abnormal uterine cell proliferation in *Cfp1*[d/d] mouse uterus (Fig. 3). They all indicate that disturbed P$_4$–PGR–SSP signaling pathway is the main cause of complete failure of embryo implantation and decidualization in *Cfp1*[d/d] mice. In fact, mice without *Gata2*, *Sox17*, or *Ihh* phenocopied all these uterine defects observed in *Cfp1*[d/d] mice[13,33,34]. In addition, SAG restored decidualization and abnormal uterine cell proliferation in *Ihh* cKO mice[33]. PGR is decreased in *Gata2*-deficient uteri, but it is necessary to mention that mice deficient in *Sox17*, *Ihh*, or *Cfp1* had normal PGR expression. Although reduced PGR expression

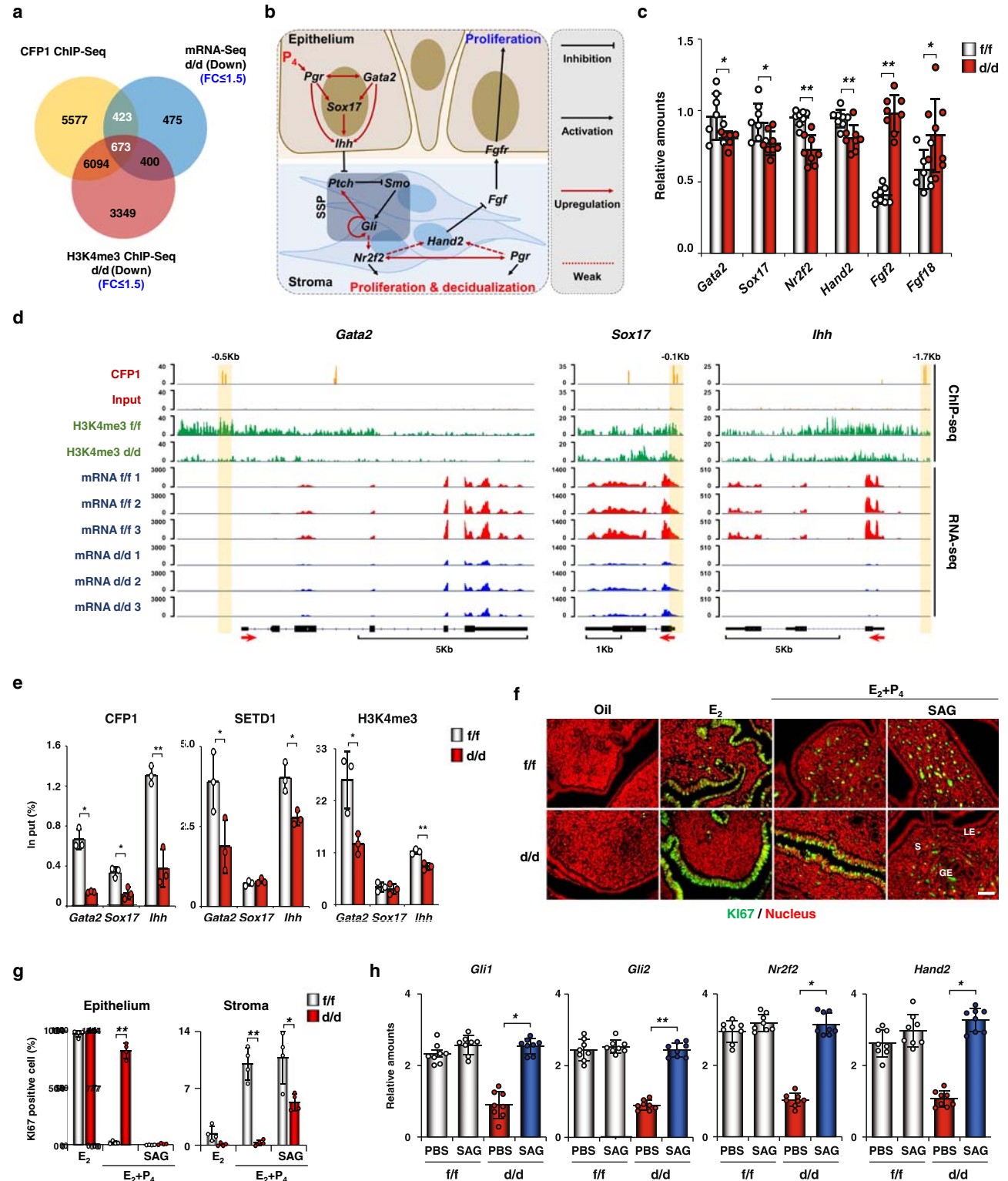

appears to be the main cause of $P_4$ resistance that contributes to the pathogenesis of endometriosis[29,38,55], some studies have reported comparable levels of PGR in eutopic[36,38] or ectopic[56] endometria of women with endometriosis. This suggests that the epigenetic aberration in PGR downstream pathways could be involved in the pathogenesis of endometriosis[57]. Essentially, the activation of PGR downstream pathways with SAG rescued the abnormal epithelial proliferation in $Cfp1^{d/d}$ uterus (Fig. 3) and suppressed the ectopic growth of $Cfp1^{d/d}$ uterine lesions with $P_4$ resistance in the endometriosis model

(Fig. 4). Aberrant epithelial proliferation was also found in the endometrium of patients with endometriosis[25,58]. Considering impaired $P_4$-PGR signaling without PGR reduction (Supplementary Fig. 4) in $Cfp1^{d/d}$ mice reflects $P_4$ resistance, disturbed CFP1-associated H3K4me3 could contribute to $P_4$ resistance that often leads to endometriosis in humans.

CFP1 mainly works to increase the expression of target genes through H3K4me3-dependent manners[7,52]. Accordingly, 40.3% of DEGs in $Cfp1^{d/d}$ uteri were downregulated with reduced H3K4me3

**Fig. 3 | CFP1-dependent epigenetic regulation intervenes in important uterine P₄ responses via H3k4me3-dependent and H3k4me3-independent manners in mice. a** Venn diagram of identifying genes that overlapped in datasets between CFP1-binding sites on their extended gene bodies (2 Kbp upstream and 200 bp downstream of TSS and CGI), downregulated gene in *Cfp1*^d/d mice (>1.5 folds), and downregulated H3K4me3 sites in *Cfp1*^d/d mice (>1.5 folds) on Day 4. **b** Schematic cartoon to show the epithelium−stroma crosstalk mediated by SSP. **c** Real-time RT-PCR analyses of upstream and downstream genes in IHH-dependent SSP (*Gata2, Sox17, Nr2f2, Hand2, Fgf2,* and *Fgf18*). *n* = 8 biologically independent samples per genotype. Data are presented as mean values with SD. Statistical analyses were performed using the unpaired Student's *t*-tests. *\*p* < 0.05, *\*\*p* < 0.01. **d** Integrative Genomics Viewer screenshots that show the distribution of CFP1 binding, H3K4me3 site, and RNA expression intensity in *Gata2, Sox17,* and *Ihh* of *Cfp1*^f/f and *Cfp1*^d/d mouse uterus on Day 4. **e** Real-time ChIP PCR for detecting CFP1, SETD1, and H3K4me3, binding on *Gata2, Sox17,* and *Ihh* in the uterus of *Cfp1*^f/f and *Cfp1*^d/d mice

on Day 4. *n* = 3 biologically independent samples per genotype. Data are presented as mean values with SD. Statistical analyses were performed using the unpaired Student's *t*-tests. *\*p* < 0.05, *\*\*p* < 0.01. **f** Immunofluorescent staining of KI67 in the uterus of the OVX *Cfp1*^f/f and *Cfp1*^d/d mice treated with E₂ and/or P₄ for 24 h. At 3 h after E₂ or E₂ + P₄, one uterine horn was injected with SAG and the other one with PBS. Green and red indicate KI-67 and the nucleus, respectively. S stroma, GE glandular epithelium, LE luminal epithelium. Scale bar, 50 μm. **g** Percentages of KI67-positive cells in (**f**). *n* = 4 biologically independent samples per genotype. Data are presented as mean values with SD. Statistical analyses were performed using the unpaired Student's *t*-tests. *\*p* < 0.05, *\*\*p* < 0.01. **h** Real-time RT-PCR analyses of SSP genes (*Gli1, Gli2, Nr2f2,* and *Hand2*) 3 h after SAG injection. *n* = 8 biologically independent samples per genotype. Data are presented as mean values with SD. Statistical analyses were performed using the multiple comparisons. *\*p* < 0.05, *\*\*p* < 0.01.

levels (Fig. 2d). However, a substantial portion of downregulated DEGs had even higher H3K4me3 levels (18.2%). Although CFP1 is an evolutionarily conserved epigenetic regulator to work with SETDs for H3K4me3 from yeast to mammals, recent studies have suggested that CFP1 could collaborate with epigenetic modulators other than COMPASS (complex associated with SET1) complexes, such as DNA methyltransferases (DNMTs) and HDACs, in a context-dependent manner. When CFP1/SETD1 or MLL1/2 binds to DNA, they hamper DNA methylation by blocking the access of DNMT3A[59]. CFP1 interacts with and recruits DNMT1 to suppress aberrant transcription re-initiation or silence specific genes[2,60]. The expression of *Dnmt1* and *Dnmt3a* increased gradually in the mouse uterus during early pregnancy[61]. They suggest that the H3K4me3-independent gene expression in *Cfp1*^d/d uteri could be associated with reduced DNA methylation. In addition to DNMTs, CFP1 could interact with HDAC1/2 complexes to regulate fertility and development in *C. elegans*. CFP1 recruits the HDAC complex to H3K4me3-rich promoter regions to deacetylate chromatin[62]. CFP1-dependent H3K4me3 is necessary to recruit histone acetylase(s) for H3K9ac dynamics in mouse embryonic stem cells[6], suggesting that CFP1-associated H3K4me3 cross-talks with histone acetylation. Interestingly, higher P₄ levels in in vitro fertilization cycles on the day of hCG administration altered various epigenetic marks, including H3K9ac, H3K9me2, and H3K27me3, in the endometrium[63], suggesting that histone modifications and P₄-PGR signaling influence each other in the endometrium. The epithelial cells in endometriotic lesions expressed a higher level of EZH2, the enzyme responsible for a repressive mark H3K27me3, which P₄ upregulates[64]. MLL1 is directly regulated by P₄–PGR signaling in the uterus and MLL1 and H3K4me3 both decreased in the eutopic endometrium of patients with endometriosis[65]. Furthermore, the expression of HDAC3, one of HDACs that reduce gene expression, was significantly lower in the endometrium of patients with endometriosis. The loss of HDAC3 in mice leads to infertility that results from embryo implantation failures with defective decidualization possibly through the aberrant activation of *Col1a1* and *Col1a2* genes that promote fibrosis with decreased ESR and PGR in the uterus[25]. In summary, our results suggest that aberrant epigenetic regulation in CFP1-deficient mice provides a uterine environment with P₄ resistance that leads to infertility caused by multiple failures and endometriosis (Fig. 6). This study is of great significance to provide an underlying epigenetic mechanism of P₄ resistance in the endometrium that could lead to endometriosis in humans.

## Methods
### Animals
All mice used in this study were maintained in accordance with the policies of the CHA University Institutional Animal Care and Use Committee (No150083). Adult (8–10 weeks of age) C57BL/6 mice, provided by Orient Bio (Gapyeong, Gyeonggi, Korea), were housed

under temperature- and light-controlled conditions with the light on for 12 h daily and fed ad libitum. *Cfp1*^f/f mice were kindly provided by Dr. David G. Shkolnik's laboratory[1]. First, *Cfp1*^f/f mice were mated to *Pgr*^cre/+ mice to generate *Cfp1*^f/+;*Pgr*^cre/+ mice[66]. Then, these mice were crossed to generate *Cfp1*^d/d and *Cfp1*^f/f mice. Genotyping PCR was performed by genomic DNA extracts from tail biopsies (Supplementary Table 3).

### Vaginal smear analysis and fertility test
Estrous cyclicity was evaluated in mature *Cfp1*^f/f and *Cfp1*^d/d female mice by daily analysis of vaginal smears over 2 weeks between 8:00 and 9:00 AM. The estrous cycle stage (proestrus, estrus, metestrus, and diestrus) was determined based on the presence of vaginal cornified epithelial cells and nucleated epithelial cells/total vaginal cells. To evaluate reproductive performance, mature *Cfp1*^f/f and *Cfp1*^d/d female mice (*n* = 4 per genotype) were individually bred to wildtype males with proven fertility. The numbers of litters and pups were recorded for 2 months.

### Serum E₂ and P₄ level measurement
Blood samples were collected on Day 4 (9:00 AM) from the heart of *Cfp1*^f/f and *Cfp1*^d/d female mice (*n* = 5 to 9 per genotype). First, the mouse blood was sampled without anticoagulant and transferred to a sterile empty tube. Next, the mouse blood was centrifuged at 1500 × *g* for 15 min at 4 °C. Separated serum was transferred into a new sterile empty tube. Serum levels of mouse E₂ and P₄ were measured by radioimmunoassay[67].

### Ovulation, fertilization, and preimplantation embryo development
*Cfp1*^f/f and *Cfp1*^d/d female mice were bred to wildtype males with known fertility. The morning of the vaginal plug observation was designated as Day 1. The mice were sacrificed on Day 2, and their oviducts were flushed to evaluate the number and fertilization potential of ovulated oocytes (*n* = 7 per genotype). In addition, fertilized 2-cell embryos were cultured up to the blastocyst stage in 20 ml droplets of KSOM (Millipore, Danvers, MA, USA) covered with oil (SAGE In-Vitro Fertilization, Inc., Trumbulla, CT, USA) in a petri dish.

### Hormone treatments
To investigate time-dependent actions of E₂ or P₄ on the expression of *Cfp1* in C57BL/6 mice uterus, adult (8–10 weeks of age) female mice were OVX, rested for 14 days, and then subcutaneously injected with either vehicle (sesame oil, 0.1 mL/mouse; Acros, NJ, USA), E₂ (100 ng/mouse, Sigma–Aldrich, St. Louis, MO, USA) or E₂ + P₄ (2 mg/mouse, Sigma–Aldrich). After hormone injection, the mice were sacrificed at various time points (0–24 h) and the uterus was collected for real-time RT-PCR (*n* = 4 to 6 per each group).

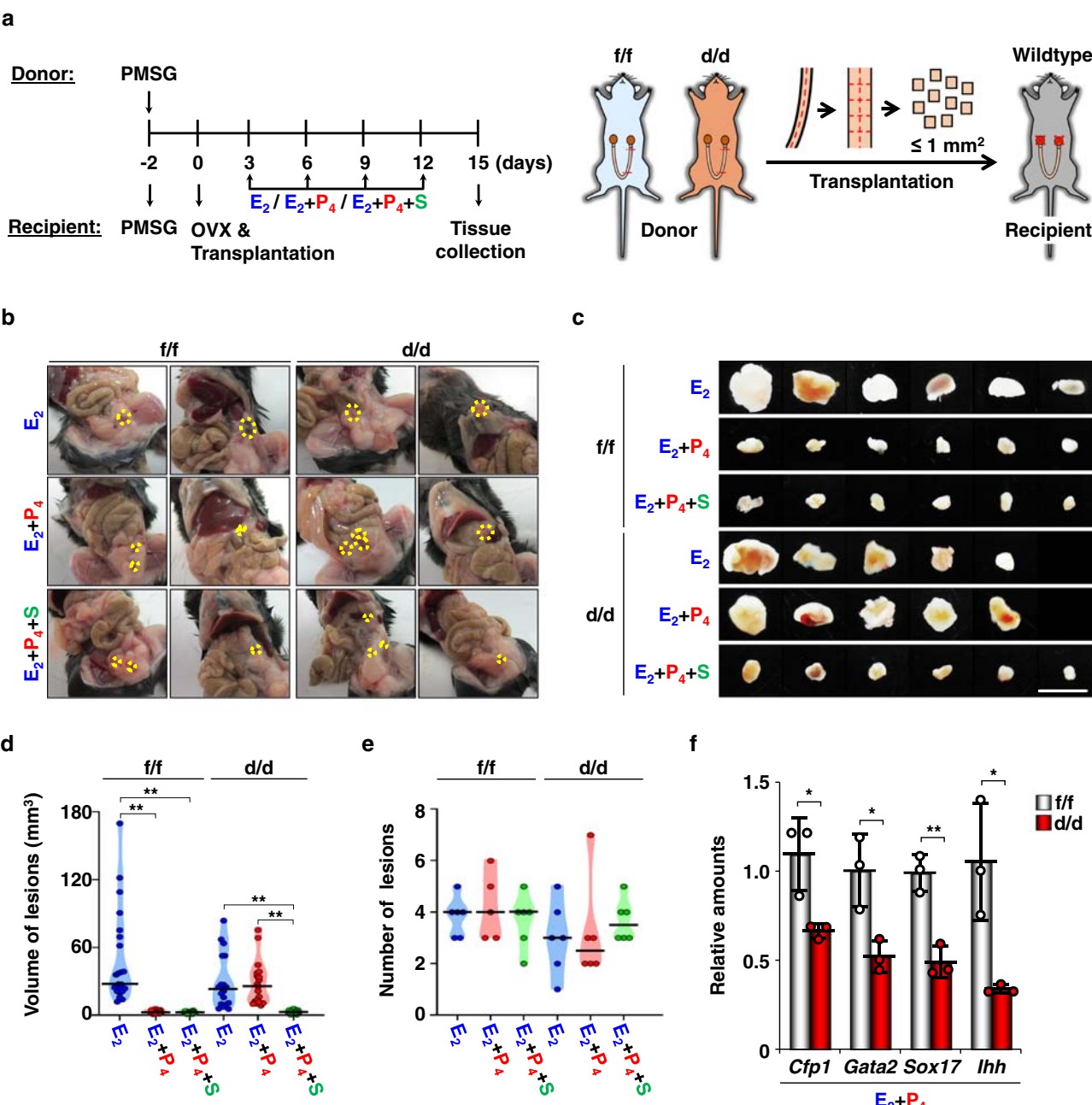

**Fig. 4 | CFP1 is required for the counteraction of P₄ on E₂-dependent uterine growth in a mouse model of human endometriosis. a** Schematic protocol to induce an experimental model of human endometriosis with transplantation of *Cfp1*[f/f] or *Cfp1*[d/d] uterine fragments. **b**–**e** Representative photographs to show the in vivo locations (**b**) and size (**c**) of ectopic lesions (dashed circles) harvested 15 days after transplantation of small pieces of *Cfp1*[f/f] and *Cfp1*[d/d] uterine tissues in recipients treated with E₂, E₂ + P₄ + PBS, or E₂ + P₄ + SAG. Scale bar, 5 mm. Average volumes (**d**) and numbers (**e**) of ectopic lesions collected from recipients. *n* = 5 to 6 biologically independent samples per group. Statistical analyses were performed using the multiple comparisons. *$p < 0.05$, **$p < 0.01$. **f** Real-time RT-PCR analyses of *Cfp1*, *Gata2*, *Sox17*, and *Ihh* expression in *Cfp1*[f/f] and *Cfp1*[d/d] ectopic uterine lesions in recipients treated with E₂ and P₄. *n* = 3 biologically independent samples per genotype. Data are presented as mean values with SD. Statistical analyses were performed using the unpaired Student's *t*-tests. *$p < 0.05$, **$p < 0.01$.

## Tissue preparation

Female reproductive organs under various conditions, such as early pregnancy, ovarian steroid hormone treatment, and artificial decidualization, were dissected and then fixed in 4% paraformaldehyde. Fixed tissues were washed, dehydrated, and embedded in paraplast (Merck KGaA, Darmstadt, Germany). Paraffin-embedded tissues were sectioned to 5 μm thickness using a microtome, stained with hematoxylin and eosin (Sigma–Aldrich), and observed by light microscopy.

## Early pregnancy and embryo implantation

Pregnancy was evaluated by the presence of a vaginal plug on the next morning after breeding with a fertile male. The pregnant mice were sacrificed on various days of pregnancy from Day 1 to 5, and their uterine horns were collected and processed for the following experiments (*n* = 4 to 6 per each group). IS were detected by intravenous (i.v.) injection of 1% Chicago Sky Blue (in saline) on the morning of Day 5, and the number of IS demarcated by blue spots was recorded[68].

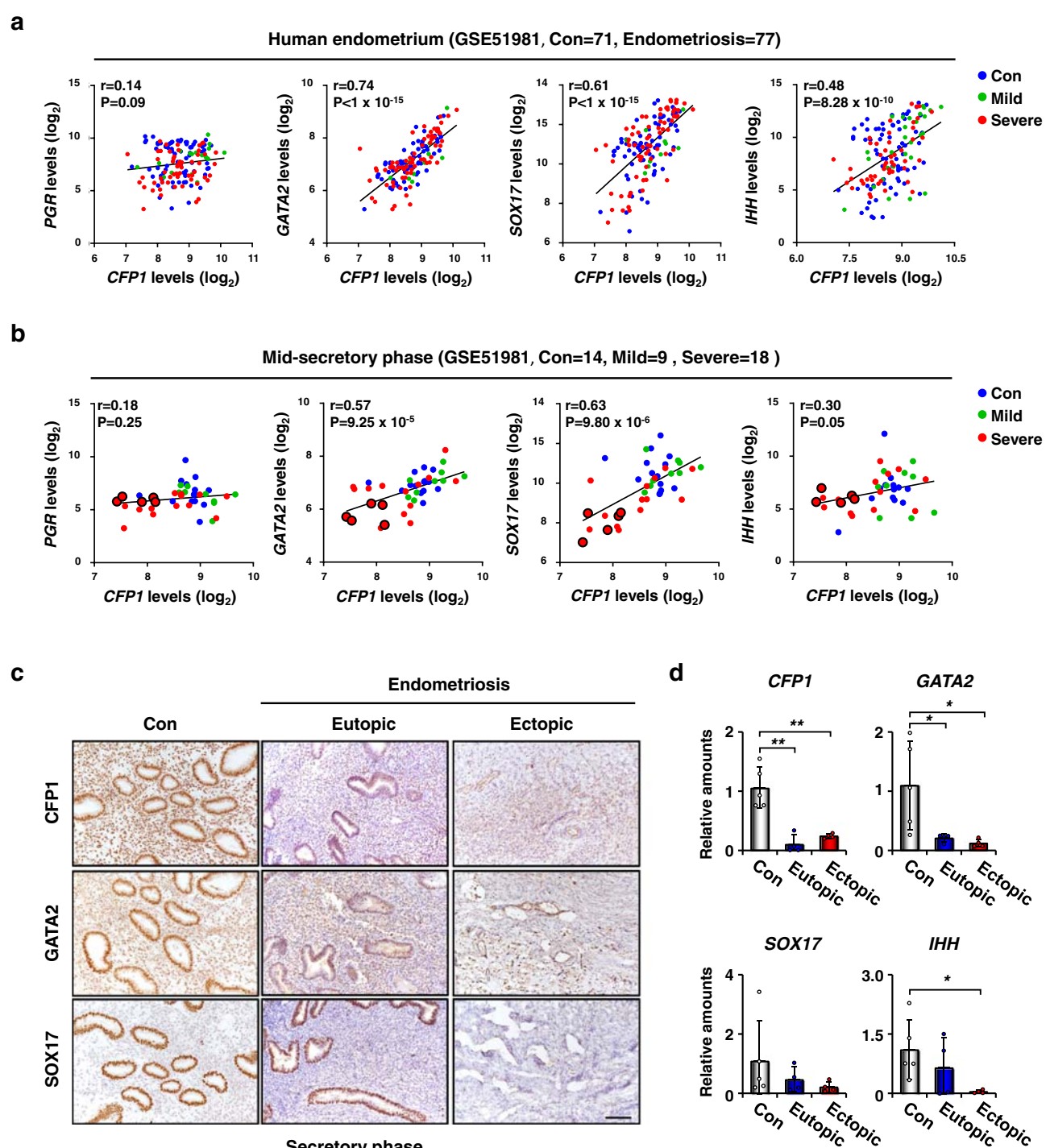

**Fig. 5 | Concomitant reduction of *CFP1* and important P₄ responsive genes observed in the endometrium of patients with endometriosis. a** Scatterplot of the positive correlations between mRNA expression levels of *CFP1* and important P₄-responsive genes, *GATA2*, *SOX17*, and *IHH*, in human endometrium of healthy women (control, blue dots) and patients with mild (green) or severe (red) endometriosis (data from GSE51981). **b** Positive correlations were observed between the expression levels of *CFP1* and *GATA2*, *SOX17*, and *IHH* in the human endometrium in the med-secretory phase in (**a**). Note that a subgroup of patients with severe endometriosis had lower expression of *CFP1* and important P₄-responsive genes without *PGR* reduction (circled red dots). r indicates Pearson correlation coefficient. **c–d** Immunohistochemical staining (**c**) and real-time RT-PCR (**d**) of *CFP1*, *GATA2*, *SOX17*, and/or *IHH* in the human endometrium of healthy women (control) and patients with severe endometriosis (eutopic and ectopic) on the secretory phase. *n* = 4 to 5 biologically independent samples per group. Scale bar, 100 μm. Data are presented as mean values with SD. Statistical analyses were performed using the multiple comparisons. *p < 0.05, **p < 0.01.

## Embryo transfer

Embryo transfer was performed as previously described with some modifications[69]. Four-week-old C57BL/6 mice were given intraperitoneal (i.p.) injections of 5 IU PMSG (Sigma–Aldrich) followed by i.p. injections with 5 IU hCG (Sigma–Aldrich) and then mated with fertile male mice to obtain the embryos for embryo transfers. The blastocysts were transferred to the uteri of either pseudopregnant day 4 *Cfp1*^f/f and *Cfp1*^d/d recipient mice, which were mated with vasectomized C57BL/6

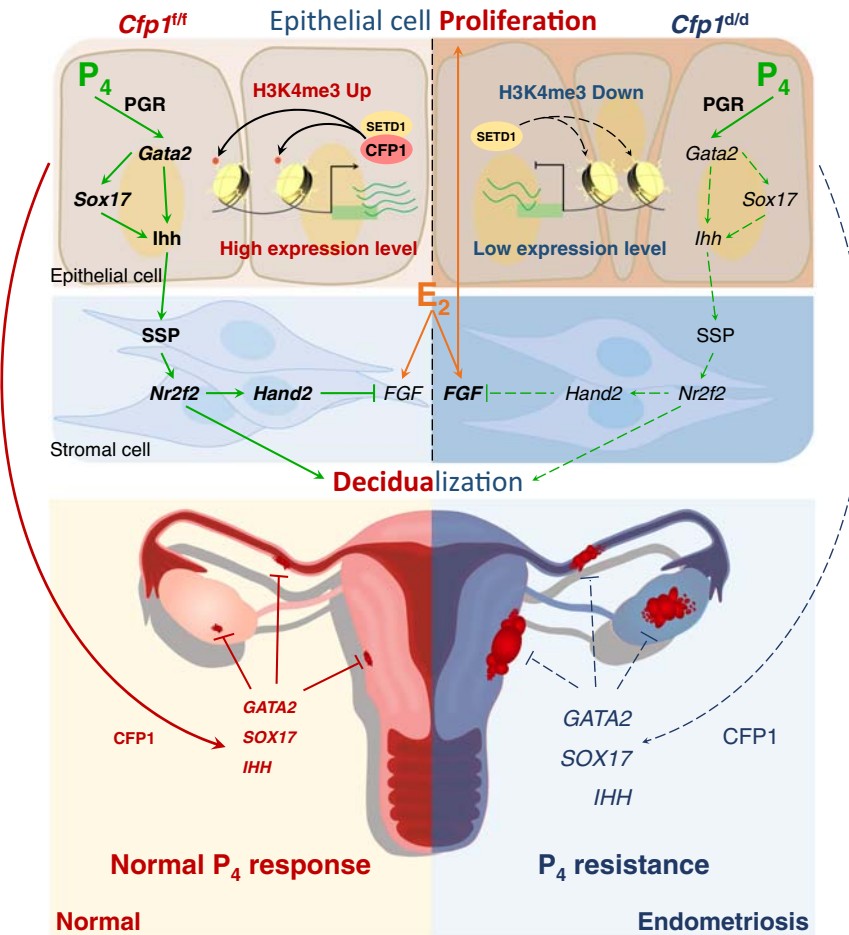

**Fig. 6 | A schematic illustration of CFP1 function in P$_4$-epigenome-transcriptome networks for uterine physiology.** CFP1 function to maintain P$_4$-PGR signaling for embryo implantation during early pregnancy and suppression of endometriosis in the uterus.

male mice. Embryo implantation was evaluated by an i.v. injection (0.1 mL/mouse) of Chicago Sky Blue (1% in saline, Sigma–Aldrich) 24 h after embryo transfer.

### Artificial decidualization

Artificial decidualization was experimentally induced, as previously described, with some modifications[69]. Adult (8–10 weeks of age) *Cfp1*$^{f/f}$ and *Cfp1*$^{d/d}$ female mice were OVX and rested for 14 days and then received daily injections of 100 ng E$_2$ for 3 days (*n* = 3 to 4 per genotype). After 2 days of resting, the mice were then treated with daily injections of 1 mg P$_4$ and 10 ng E$_2$ for 3 days. At 6 h after the last injection, one uterine horn was traumatized by the injection of 50 μL of sesame oil. Mice were given daily injections of P$_4$ (1 mg) + E$_2$ (10 ng) following trauma. The weight of stimulated and non-stimulated uterine horns was recorded 4 days after the oil infusion. The fold increase in uterine weights and alkaline phosphatase staining served as an index of decidualization.

### Intrauterine delivery of SAG

SAG (Abcam, Cambridge, UK), a smoothened agonist, was applied to rescue aberrant uterine cell proliferation and gene expression in *Cfp1*$^{d/d}$ female mice[33]. Adult (8–10 weeks of age) *Cfp1*$^{f/f}$ and *Cfp1*$^{d/d}$ female mice were OVX and rested for 14 days. Each OVX mouse was treated with 2 mg P$_4$ for 2 days and then 2 mg P$_4$ and E$_2$ 60 ng/mouse on the third day. At 3 h after E$_2$ and P$_4$ injection, 10 μL of vehicle (PBS) and SAG (1000 nM) in PBS were injected intraluminally in each horn of the uterus. Mice were sacrificed at 3 or 21 h after SAG injection depending on experimental conditions, and the uterus was collected

for real-time RT-PCR and immunofluorescence staining (*n* = 8 per each group).

### RNA extraction, RT-PCR, and real-time RT-PCR

Total RNA was extracted from mouse uteri under various conditions (*n* = 3 to 8 per each group) using Trizol Reagent (Invitrogen Life Technologies, San Diego, CA, USA) according to the manufacturer's protocols. The first-strand cDNA was synthesized from 1 μg of total RNA using M–MLV reverse transcriptase (Promega, Madison, WI, USA) and RNasin Ribonuclease inhibitor (Promega). Synthesized cDNA (10 ng) was utilized for PCR with specific primers at optimized cycles (Supplementary Table 4). To quantify expression levels, real-time RT-PCR was performed using the SYBR green dye (iQ SYBR Green Supermix, Bio-Rad, Hercules, CA, USA), as previously described[68]. To compare transcript levels between samples, a standard curve of cycle thresholds for several serial dilutions of a cDNA sample was established and then used to calculate the relative abundance of each gene. Then, values were normalized to the relative amounts of *Rpl7* cDNA. All reactions were performed in duplicates.

### Western blotting

Uterus tissues (*n* = 4 to 6 per each group) were lysed in lysis buffer including PRO-PREP (iNtRON, Seongnam, Korea) solution and 1X phosphatase inhibitor (Roche Applied Sciences, Indianapolis, IN, USA). The protein samples (20 μg/lane) were then separated by 8% SDS-PAGE, transferred onto nitrocellulose membrane (Bio-Rad), and blocked with 5% non-fat milk (Bio-Rad) in TBS (Bio-Rad) containing 0.1% Tween 20 (Sigma–Aldrich). Antibodies used for Western blotting,

immunostaining, and/or ChIP were summarized in Supplementary Tables 5 and 6. The signals were developed using an ECL Western blotting substrate kit (Bio-Rad) and detected using a Chemidoc XRS+ (Bio-Rad) with Image Lab software.

### Immunohistochemistry and immunofluorescence analysis

Paraffin-embedded tissues were sectioned to 5 μm thickness using a microtome. Uterine sections were deparaffinized and rehydrated. Endogenous peroxidase was inactivated with 3% $H_2O_2$. Sections were subjected to antigen retrieval in 0.01 M sodium citrate buffer (pH 6.0). Nonspecific staining was blocked using protein block serum (Dako, Carpinteria, CA, USA) for 1 h. Sections were incubated with primary antibodies at 4 °C overnight. On the following day, sections were incubated with appropriate secondary antibodies for 1 h at room temperature. Sections were counterstained using Topro-3-iodide (TOPRO; Life Technologies, Carlsbad, CA, USA) and mounted. For immunohistochemistry, DAB reagent (Vector Laboratories, Inc., Burlingame, CA, USA) was applied to visualize signals. Images were observed under a microscope (Carl Zeiss, Oberkochen, Germany) and analyzed using ZEN software (Carl Zeiss).

### mRNA-seq and data analysis

Libraries were prepared from 2 μg of total RNA in $Cfp1^{f/f}$ and $Cfp1^{d/d}$ mouse uterus on Day 4 using the SMARTer stranded mRNA-Seq kit (Clontech Laboratories, Inc., USA). mRNAs were used for the cDNA synthesis and shearing, following the manufacturer's instructions ($n = 3$ pools: 3 mice per each pool). The Illumina indexes were used, and the enrichment step was conducted with PCR. The RT (read count) data were processed based on the quantile normalization method using EdgeR within R[70] utilizing bioconductor. The alignment files were also used for assembling transcripts, estimating their abundances, and detecting differential expression of genes or isoforms using cufflinks. These are performed using Bowtie2 software. We also used the fragments per kilobase of exon per million (FPKM) fragments to determine the expression levels of the gene regions. Gene classification was based on searches made by GSEA, and heatmaps of the unsupervised hierarchical clustering and DEGs were produced using MeV software.

### CFP1 immunoprecipitation

Immunoprecipitation of CFP1 protein using CFP1 antibodies was performed with the manufacturer's instructions using immunoprecipitation kit (Abcam). In addition, CFP1 Western blotting following immunoprecipitation was performed to evaluate the specificity of CFP1 antibodies used for CFP1 ChIP-seq for $Cfp1^{f/f}$ uterine samples (Supplementary Fig. 10). On Day 4, uterine horns of $Cfp1^{f/f}$ and $Cfp1^{d/d}$ mice were cut vertically, and epithelial and stromal cells were separated from the smooth muscles ($n = 4$ to 5 per each group). Then, tissues were incubated in a lysis buffer with protease inhibitors and mixed on a rocker at 4 °C for 1 h. The tissue extracts were transferred to a fresh tube, and a predetermined amount of antibodies was added on a rocker at 4 °C for 4 h. After antibody binding, protein A/G Sepharose beads were added on a rocker at 4 °C for 1 h. Next, protein A/G Sepharose beads were collected, washed, and eluted. The protein of interest was purified by low-speed centrifugation at 4 °C and used for Western blotting.

### ChIP and real-time ChIP PCR

ChIP analysis was performed with a slight modification of the manufacturer's instructions using the ChIP-IT Express Enzymatic kit (Active Motif, Carlsbad, CA, USA). On Day 4, $Cfp1^{f/f}$ and $Cfp1^{d/d}$ mouse uterus horns were cut, and epithelial and stromal cells separated from the uterine smooth muscles ($n = 3$ pools: 3 mice per each pool). Then, epithelial and stromal cells were fixed in DMEM high-glucose media containing 1% formaldehyde and then made into single cell slurry

through an electric homogenizer. DNA fragments between 150 and 500 bp were obtained by enzymatic shearing cocktail after lysis cell and nuclei isolation through lysis buffer and Dounce homogenizer. Antibodies were added in sheared chromatin, and immunoprecipitation was performed overnight at 4 °C. Immunoprecipitated DNA was utilized for real-time ChIP PCR with specific primers (Supplementary Table 1). To quantify the enrichment level in ChIP-seq data, real-time ChIP PCR was performed with iQTM SYBR Green Supermix (Bio-Rad) on BIO-RAD iCycler using immunoprecipitated DNA. All PCR reactions (10% input, CFP1, SETD1, H3K4me3, and normal IgG) were duplicated ($n = 3$ to 4 per each group).

### ChIP-seq and data analysis

The library was constructed using NEBNext® UltraTM DNA Library Prep Kit for Illumina (New England Biolabs, UK) according to the manufacturer's instructions. Briefly, the chopped DNA was ligated with adaptors. After purification, the PCR reaction was conducted with adaptor-ligated DNA and index primer for multiplexing sequencing. The library was purified using magnetic beads to remove all reaction components. The library size was assessed by Agilent 2100 bioanalyzer (Agilent Technologies, Amstelveen, Netherlands). High-throughput sequencing was performed as paired-end 100 sequencing using HigSeq 2500 (Illumina, Inc. USA). ChIP-seq metaplots and heatmaps were analyzed using DeepTools software.

### Experimentally induced mouse model of endometriosis

Recipient (wild type) and donor ($Cfp1^{f/f}$ and $Cfp1^{d/d}$) female mice (8–10 weeks of age) were primed with 5 IU PMSG (Sigma–Aldrich) for 48 h to stimulate uterine growth. After inhalation anesthesia, the uterine horn was collected from donor mice and vertically opened with scissors in a petri dish containing warmed Dulbecco's phosphate-buffered saline (37 °C). The uterine horn was cut into small pieces of approximately 1 mm² and injected into the peritoneum of the OVX recipient mice. After transplantation of uterine tissues in recipient mice, $E_2$ (100 ng/mouse), $P_4$ (2 mg/mouse), PBS, and/or SAG were injected every 3 days. At 15 days after transplantation, the volume and number of ectopic lesions were measured, and ectopic lesions were collected for real-time RT-PCR.

### Human endometrial sampling

Control ($n = 7$) and endometriotic endometrial tissues ($n = 5$) were collected from patients who underwent hysteroscopy–laparoscopy surgery to evaluate endometrial abnormalities, including endometriosis in the Department of Obstetrics and Gynecology of the First Affiliated Hospital of Xiamen University. The sample collection and studies reported here were approved by the ethics committee of Hospital of Xiamen University (XMYY-2021KYSB044), and all participants signed the informed consent. Endometriotic lesions were obtained from women (aged 25–35 years) suffering from ovarian endometriosis, confirmed by laparoscopy and histopathology. These patients had regular menstrual cycles and were recruited without hormone treatment for at least 3 months before surgery. All samples were collected during the luteal phase of the menstrual cycle.

### Statistical analysis

GraphPad Prism version 8 software (GraphPad Software, La Jolla, CA, USA) was used for statistical analyses. All values are represented as mean ± standard deviation. Statistical analyses were performed using the unpaired Student's $t$-tests, and $p < 0.05$ was considered statistically significant.

### Reporting summary

Further information on research design is available in the Nature Portfolio Reporting Summary linked to this article.

## Data availability

Raw data files are deposited in the NCBI Gene Expression Omnibus database under Super Series accession number GSE219104. The Supplementary Fig. 7 data re-analyzed in this study are available in the GSE118264 database[13], GSE40661 database[34], and GSE178541 database[35]. All other data supporting the findings of this study are available within the paper and its Supplementary Information. Source data are provided with this paper.

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

## Acknowledgements

This research was supported by Basic Science Research Program through the National Research Foundation of Korea (NRF) funded by the Ministry of Education (NRF- 2019R1A6A1A03032888 to H.S.), by the NRF grant funded by the Korea government (the Ministry of Science and ICT) (NRF-2015R1A2A2A01006714 to H.S., NRF- 2020R1C1C1008317 to H.R.K., NRF-2020R1A2C2005012 to H.S., and NRF-2020R1A6A3A01100338 to M.P.), by Bio & Medical Technology Development Program of the NRF and funded by the Korean government (MSIT) (NRF-2022M3A9E4016936 to S.-H.H.).

## Author contributions

S.C.Y. and H.S. conceived and designed the experiments. S.C.Y., M.P., H.L., P.W., C.P., G.L., Q.C., S.H.P., and Y.S.K. performed experiments. S.C.Y., M.P., K.-H.H., Y.C., F.J.D., J.P.L., D.G.S., H.J.L., S.-H.H., H.-R.K. and H.S. analyzed the results. S.C.Y., M.P., and H.S. discussed the results and wrote the manuscript.

## Competing interests

The authors declare no competing interests.

## Additional information

[1]Department of Biomedical Science, CHA University, Seongnam, Gyeonggi 13488, Korea. [2]Department of Stem Cell and Regenerative Biotechnology, Konkuk University, Seoul 05029, Korea. [3]Fujian Provincial Key Laboratory of Reproductive Health Research, Department of Obstetrics and Gynecology, The First Affiliated Hospital of Xiamen, School of Medicine, Xiamen University, Xiamen, Fujian 361102, China. [4]Department of Reproductive and Developmental Biology Laboratory, National Institute of Environmental Health Sciences, Research Triangle Park, NC 12233, USA. [5]Department of Molecular and Cellular Biology and Center for Reproductive Medicine, Baylor College of Medicine, Houston, TX 77030, USA. [6]Department of Biology, School of Science, Indiana University-Purdue University Indianapolis, Indianapolis, IN 46202, USA. [7]Department of Veterinary Science, Konkuk University, Seoul 05029, Korea. [8]Department of Internal Medicine, School of Medicine, Kangwon National University, Chuncheon, Gangwon-do 24431, Korea. [9]KW-Bio Co., Ltd, Wonju 26493, Korea. [10]These authors contributed equally: Seung Chel Yang, Mira Park. ✉e-mail: hssong@cha.ac.kr

