## [Peer Review File · Nature Communications]

CFP1 governs uterine epigenetic landscapes to intervene in progesterone responses for uterine physiology and suppression of endometriosisREVIEWER COMMENTS

Reviewer #1 (Remarks to the Author):

This study investigates a role of CFP1 on progesterone (P4) resistance using *Cfp1*^{flox/flox};*Pgr*-Cre (*Cfp1*^{d/d}) mice. *Cfp1*^{d/d} mice reveal several reproductive phenotypes in ovary, uterus and endometriosis model. The mutant mice are infertile due to implantation failure and non-receptive endometrium. Transcriptomic and ChIP-seq analysis identify P4 responsive genes including *Gata2*, *Sox17*, and *Ihh* as direct CFP1 targets. Interestingly, some phenotypes of mutant mice are rescued by a smoothed agonist. Furthermore, they confirm that CFP1 is significantly downregulated in women with endometriosis compared to control and expression levels between CFP1 and these P4 targets are positively related regardless of PGR levels. Based on strong results, they conclude that CFP1 is a critical epigenetic regulator for P4 signaling on uterine functions. The study is carefully designed and meticulously executed. The manuscript is well written in a clear manner and results support the conclusion. The findings are interesting and provide some insight into the regulation of progesterone signaling during pregnancy. Please clarify the following points for revision. However, there are some points to be addressed.

1. This study shows that CFP1 is a critical epigenetic regulator for P4 response using *GATA2*, *SOX17*, and *IHH* genes. There are a lot of P4 responsive genes. Have you compared your transcriptomic data to other P4-related transcriptomic data sets? It will give us more confident conclusion.
2. Strong CFP1 expressions are detected in ovary. Do *Cfp1* ^{d/d} mice have abnormal ovarian functions including ovulation, fertilization, and ovarian hormone production?
3. Supplementary Fig. 2C: There are several CFP1 expressed cells in the uterus. Especially, uterine epithelial cells show high CFP1 expression in *Cfp1* ^{d/d} mice. Is there inefficient ablation of *Cfp1* in the mutant mice?
4. Fig. 3H: Are the recovery rates of P4 target genes by SAG compatible to wild type mice? Add wild type or *f/f* control mice treated with PBS and SAG to the rescue experiment.
5. Line 5: Rewrite the sentence. There is strong relationship between infertility and endometriosis. However, we have not known the mechanisms including progesterone resistance.
6. Line 19; Provide specific conditions for physiologic and pathophysiologic conditions.

Reviewer #2 (Remarks to the Author):

I congratulate the authors for this manuscript. I believe their findings will help advance our

understanding of both the uterine epigenetic machinery and in the pathophysiology of endometriosis. They have shown in the *Cfp1^{flox/flox};Pgr-Cre* mouse model that while *cfp1* is absent in most cells of mouse adult FRT, this does not affect the morphology of cells yet normal decidualization, blastocyst implantation, implantation sites and live pups were aberrant, indicating a deficient uterine microenvironment. They then provide further evidence on the P4-CFP1 interplay both dependent and independent of H3K4me3, and via IHH-dependent pathway for this aberrancy. They examined both stromal and epithelia cells and by using a Smo agonist.

For potential role of CFP1 in endometriosis they use a mouse model to assess P4-response via CFP1 in ectopic lesion growth. Evaluating GATA2, SOX17, and IHH mRNA levels (and CFP1 and PGR) in a human mild, severe and non-disease dataset they suggest a potential role of CFP1 in suggested P4-resistance in endometriosis.

1) I do agree that their data show an important role for CFP1 in aberrant epigenetic landscape in patients with endometriosis but, both their mouse model and the human subset they have evaluated are models of existing lesions and does not provide evidence for a “leading” role for CFP1 (as they have suggested in the end of the Results section). This should be re-worded. Why and how some women develop endometriosis, from immune evasion, to attachment in the peritoneum/on organs, to sustained and expanded growth, the potential role of CFP1 in each of these are yet to be determined, and we cannot yet assume a leading role for CFP1 in endometriosis.

Besides the above point, the following few points are needed to be addressed to ensure optimum accuracy of the presented manuscript:

2. In Figure 2A the authors have CFP1 ChIP-Seq for f/f mice, but also need to include for d/d mice (no IP would be expected, but it is necessary to show for comparison). This is important because the aim of this experiment was to show that H3K4me3 was reduced (in a CFP1-dependent manner) in TSS and CGI in *Cfp1^{d/d}* mouse uterus, due to lack of CFP1.

3. Figure 2i; Add RT-PCR to the graph. Otherwise the title atop is confusing.

4) From human CFP1 data, we know that CFP1 occupies active CGI TSSs, but it is not restricted to them. Rather, it also occupies non-CGI TSSs and enhancers of actively transcribed genes. The authors have shown a significant portion of DEGs with decreased expression levels, but show increased H3K4me3 levels in *Cfp1^{d/d}* mouse on Day 4, suggesting an H3K4me3-independent manner for CFP1 gene expression. They also show that the “number of significantly downregulated (3329) and upregulated (2829) genes in *Cfp1^{d/d}* mice was comparable (54% vs. 46%) in DEGs from mRNA-seq. However, further analysis of gene ontology (GO) terms showed that 130 most GO terms (126/132, 95.5%) with false discovery rate (FDR) of <0.25 were reduced in the uteri of *Cfp1^{d/d}* mice on Day 4.”

My understanding from the authors description here, is that even though they have seen comparable

up- and down-regulation in their mRNA result, they are assuming more down-regulation than up-regulation based on their subsequent GO analysis. This interpretation may or may not be accurate. The GO algorithm, among other things, measures “representation” of terms, either over- or under-representation. This could also depend on the frequency in either an input background or sample frequencies. And may or may not reflect the underlying biology. Therefore, one could assume that not very frequent, but still significant, upregulated genes may not be “enriched” in the GO output. This would also include unresolved gene names. GO does provide these lists with the table of outputs. Note that the lack of assignment of a gene/ groups of genes by GO, in this case the upregulated genes, does not invalidate the mRNA expression data. Or their prevalence. The down-regulating genes have shown a strong enrichment in specific functions/pathways, and it is of great interest; however, up-regulated genes, may be scattered across the genome, as I mentioned above, not at CGI TTSs, each with a potentially significant biological role, but without particular sets of genes to indicated specific functions/pathways/terms. Therefore, GO is not a downstream validating analysis to the mRNA expression data and the authors cannot assume a more prominent down-regulating role on the basis of their GO analysis. They must edit the wording in the text to reflect this.

5. Line 151: “Interestingly, the upstream and downstream genes of 149 IHH-dependent SSP were included in the list; *Ihh*, *Gli3*, and *Gata2* as H3K4me3-dependent and *Ptch1*, *Sox17*, and *Nr2f2* as H3K4me3-independent target genes (Fig. 3B and Supplementary Table 1). The expression of these genes was significantly reduced in *Cfp1d/d* mice (Figs. 2I and 3C)”.

Unlike RT-PCRs in Fig 2I, in figure 3C, the relative expression change is statistically significantly changed, but not significantly. Meaning these relative expressions changes are not that significant/large, even if this reaches statistical significance. This wording must be corrected in the text for accuracy.

6. Also as in No 5, in text re Supp Fig 8B (line 204), this should be corrected to statistically significantly different. As expression levels are not that substantially different, even though reaching statistical significance.

7. I find the data on H3K4me3 on *Gata2*, *Ihh*, and *Sox17*, in *Cfp1d/d* mice and the subsequent validation of promoter enrichment of SETD1 and H3K4me3 of great interest. However, the enrichment at *sox17* is quite low in both mice models, and while it is likely that CFP1-SETD1 complex increases promoter H3K4me3 at *Ihh* and *Gata2*, I am not sure we can arrive at the conclusion, based on these data that CFP1 is the sole agent for *Sox17* expression. Please re-word this section for accuracy.

A point-by-point response to the reviewer's comments

We deeply appreciate critical comments from the reviewers. All the comments from the reviewers are critical to strengthening the excellence of our manuscript. To meet all the requests from the reviewers, we have tried our best for performing suggested experiments and revising the manuscript. Newly added and revised words were colored yellow in the revised manuscript.

Reviewer #1 (Remarks to the Author):

This study investigates a role of CFP1 on progesterone (P4) resistance using *Cfp1*^{flox/flox};*Pgr*-Cre (*Cfp1*^{d/d}) mice. *Cfp1*^{d/d} mice reveal several reproductive phenotypes in ovary, uterus and endometriosis model. The mutant mice are infertile due to implantation failure and non-receptive endometrium. Transcriptomic and ChIP-seq analysis identify P4 responsive genes including *Gata2*, *Sox17*, and *Ihh* as direct CFP1 targets. Interestingly, some phenotypes of mutant mice are rescued by a smoothed agonist. Furthermore, they confirm that CFP1 is significantly downregulated in women with endometriosis compared to control and expression levels between CFP1 and these P4 targets are positively related regardless of PGR levels. Based on strong results, they conclude that CFP1 is a critical epigenetic regulator for P4 signaling on uterine functions. The study is carefully designed and meticulously executed. The manuscript is well written in a clear manner and results support the conclusion. The findings are interesting and provide some insight into the regulation of progesterone signaling during pregnancy. Please clarify the following points for revision. However, there are some points to be addressed.

1. This study shows that CFP1 is a critical epigenetic regulator for P4 response using *GATA2*, *SOX17*, and *IHH* genes. There are a lot of P4 responsive genes. Have you compared your transcriptomic data to other P4-related transcriptomic data sets? It will give us more confident conclusion.

→ **Author response:** We appreciate a valuable comment from reviewer 1. As suggested by reviewer 1, we compared our datasets with other P₄-related transcriptomic datasets in different physiologic/experimental settings (GSE118264, GSE40661, and GSE178541) and provided comparison results as Supplementary Figure 7 in the revised manuscript. As shown in Supplementary Figure 7 in the revised manuscript, significant numbers of differentially expressed genes (DEGs) in our datasets (GSE219104) with *Cfp1*^{1^{fl}} and *Cfp1*^{1^{dl}} uterine samples are overlapped with those of DEGs in other P₄-related datasets, such as *Sox17* wildtype versus knockout uterus on Day 4 (GSE118264), *Gata2* wildtype versus knockout mice treated with P₄ (GSE40661), and *PgrA* and *PgrB* overexpression mice treated with oil or P₄ (GSE178541). We added a sentence to explain the results of comparative analyses in Supplementary Figure 7 in the revised manuscript (lines 175-178, lines 554-555, lines 923-928); "When our mRNA-seq datasets were compared with other P₄-related transcriptomic data (GSE118264, GSE40661, and GSE178541), comparative analyses showed that significant numbers of DEGs in *Cfp1*^{1^{dl}} uterus overlap with those of DEGs in other P₄-related datasets (Supplementary Fig. 7)".

2. Strong CFP1 expressions are detected in ovary. Do *Cfp1* d/d mice have abnormal ovarian functions including ovulation, fertilization, and ovarian hormone production?

→ **Author response:** We agree with reviewer 1 on this issue. Since CFP1 is also expressed in the

ovary, as mentioned by the reviewer, it is also important to examine the potential function of CFP1 in the ovary. Thus, we performed a series of experiments to examine whether the loss of CFP1 causes any aberration in the ovary as well as in the uterus. The revised manuscript includes all the data the reviewer mentioned, such as ovarian hormone levels, ovulation, fertilization, and *in vitro* development of embryos from *Cfp1^{d/d}* mice (Supplementary Figure 3D-F). We added sentences to demonstrate what we have in this Figure (lines 88-91, lines 103-104, lines 363-368, lines 370-376, lines 890-891, lines 897-903); “Furthermore, serum levels of E₂ and P₄ on Day 4 in *Cfp1^{d/d}* mice were comparable to those of *Cfp1^{f/f}* mice. We also found that *Cfp1^{d/d}* mice ovulate similar numbers of oocytes that can fertilize normally, and the fertilized embryos develop to the blastocyst stage without any aberrations *in vitro* (Supplementary Fig. 3D-F)” in the revised manuscript. Therefore, we conclude that CFP1 is expressed in the ovary and the uterus, but the loss of CFP1 mainly disturbs uterine P₄ response, but not ovary function in mice.

3. Supplementary Fig. 2C: There are several CFP1 expressed cells in the uterus. Especially, uterine epithelial cells show high CFP1 expression in *Cfp1 d/d* mice. Is there inefficient ablation of *Cfp1* in the mutant mice?

→ **Author response:** We are thankful for the thorough comments of reviewer 1 on the image in Supplementary Figure 2C. We agree with the reviewer’s comment on that image. The image may give a wrong impression that CFP1 may be inefficiently ablated by PR-Cre in *Cfp1^{d/d}* mice. Thus, to ensure that CFP1 is efficiently deleted in the uterus, we performed CFP1 immunostaining again in the *Cfp1^{d/d}* uterus during the revision period. The revised manuscript contains a new image of CFP1 immunostaining in Supplementary Figure 2C to take away the concern that CFP1 may not be efficiently deleted.

4. Fig. 3H: Are the recovery rates of P₄ target genes by SAG compatible to wild type mice? Add wild type or *f/f* control mice treated with PBS and SAG to the rescue experiment.

→ **Author response:** We deeply appreciate the critical comment of reviewer 1 on this experiment. We also think it is scientifically more relevant to include groups of *Cfp1^{f/f}* mice treated with PBS and SAG as controls for the experiment to rescue the phenotypes of *Cfp1^{d/d}* mice with SAG. According to the reviewer’s comment, we performed a new experiment to include two control groups (PBS and SAG treatment) in *Cfp1^{f/f}* mice during the revision period (n=8 per each group) and have a new Figure 3H in the revised manuscript (lines 167-170, lines 174-175, lines 428-430).

In the submitted manuscript, although we found that SAG significantly increased expression levels of these P₄ target genes, such as *Gli1*, *Gli2*, *Nr2f2*, and *Hand2*, in *Cfp1^{d/d}* uterus, we were not able to quantitatively evaluate how much SAG rescues the reduced expression levels of these genes. In the revised Figure 3H, we are now able to clearly mention that SAG supplementation can efficiently restore expression levels of the P₄ target genes in *Cfp1^{d/d}* uterus as comparable as that of *Cfp1^{f/f}* uterus with or without SAG. In addition, we can provide that SAG treatment did not have any additional effects on the P₄-dependent expression of these P₄ target genes in *Cfp1^{f/f}* uterus.

5. Line 5: Rewrite the sentence. There is strong relationship between infertility and endometriosis. However, we have not known the mechanisms including progesterone resistance.

→ **Author response:** We agree with the reviewer on this issue. Progesterone (P₄) resistance is a cause of various endometrial disorders, such as endometriosis, infertility, inflammatory disorders, and cancer. Endometriosis affects 10%–15% of women of reproductive age and is one of the major causes of female infertility. Thus, the sentence that reviewer 1 pointed out may not be a good one to describe the relationship between progesterone resistance, endometriosis, and infertility. According to the comment, we revised the sentence to make the relationship between them more straightforward (lines 5-6); “P₄ resistance is a major cause of the pathogenesis of endometrial disorders like endometriosis, often leading to infertility” in the revised manuscript.

6. Line 19; Provide specific conditions for physiologic and pathophysiologic conditions.

→ **Author response:** According to reviewer 1’s comment, we mentioned specific events for physiologic and pathophysiologic conditions in the abstract. Therefore, the revised sentence is “CFP1 is a key epigenetic factor that intervenes in the P₄-epigenome-transcriptome networks for uterine receptivity for embryo implantation and the pathogenesis of endometriosis” in the revised manuscript (lines 18-19).

Reviewer #2 (Remarks to the Author):

I congratulate the authors for this manuscript. I believe their findings will help advance our understanding of both the uterine epigenetic machinery and in the pathophysiology of endometriosis. They have shown in the *Cfp1^{flox/flox};Pgr-Cre* mouse model that while *cfp1* is absent in most cells of mouse adult FRT, this does not affect the morphology of cells yet normal decidualization, blastocyst implantation, implantation sites and live pups were aberrant, indicating a deficient uterine microenvironment. They then provide further evidence on the P4-CFP1 interplay both dependent and independent of H3K4me3, and via IHH-dependent pathway for this aberrancy. They examined both stromal and epithelia cells and by using a Smo agonist. For potential role of CFP1 in endometriosis they use a mouse model to assess P4-response via CFP1 in ectopic lesion growth. Evaluating GATA2, SOX17, and IHH mRNA levels (and CFP1 and PGR) in a human mild, severe and non-disease dataset they suggest a potential role of CFP1 in suggested P4-resistance in endometriosis.

1. I do agree that their data show an important role for CFP1 in aberrant epigenetic landscape in patients with endometriosis but, both their mouse model and the human subset they have evaluated are models of existing lesions and does not provide evidence for a “leading” role for CFP1 (as they have suggested in the end of the Results section). This should be re-worded. Why and how some women develop endometriosis, from immune evasion, to attachment in the peritoneum/on organs, to sustained and expanded growth, the potential role of CFP1 in each of these are yet to be determined, and we cannot yet assume a leading role for CFP1 in endometriosis. Besides the above point, the following few points are needed to be addressed to ensure optimum accuracy of the presented manuscript:

→ **Author response:** We absolutely agree with reviewer 2 on this issue. However, we do not think that we did provide evidence that CFP1 has a leading role in the pathogenesis of endometriosis. As mentioned by reviewer 2, endometriosis is a multifactorial disease that is driven by complex genetic, physiologic, and environmental interactions. In the submitted manuscript, we provide a clue to understanding the underlying epigenetic mechanism(s) that could cause progesterone resistance that may promote the development of endometriosis in mice and humans. Reviewer 2 got the impression that we imply that CFP1 has “a leading role” in the pathogenesis of endometriosis in humans because of the last subtitle in the Result section, “Downregulation of the epigenetic factor CFP1 may cause P₄ resistance, leading to endometriosis in humans”. We agree with reviewer 2 that the subtitle may overstate what we found in the submitted manuscript. We did not mean to state that aberrant down-regulation of CFP1 expression in human endometrium could be a leading cause of endometriosis. Thus, we tone-downed the last subtitle to “Downregulation of the epigenetic factor *CFP1* may be associated with endometriosis in humans” accordingly in the revised manuscript (lines 200-201).

2. In Figure 2A the authors have CFP1 ChIP-Seq for *f/f* mice, but also need to include for *d/d* mice (no IP would be expected, but it is necessary to show for comparison). This is important because the aim of this experiment was to show that H3K4me3 was reduced (in a CFP1-dependent manner) in TSS and CGI in *Cfp1^{d/d}* mouse uterus, due to lack of CFP1.

→ **Author response:** We absolutely agree with reviewer 2 on this issue. We also had the same concern that reviewer 2 mentioned from the very beginning. Initially, we were going to send not only *Cfp1^{f/f}* but also *Cfp1^{d/d}* uterine samples for CFP1 ChIP-Seq. However, the company that performed the sequencing with our samples suggested going with only *Cfp1^{f/f}* uterine samples. The reason is that the library for ChIP-Seq can

be constructed with the DNA segments precipitated by CFP1 antibodies in *Cfp1^{d/d}* uterine cells but not in *Cfp1^{d/d}* uterine cells since CFP1 protein barely exists in *Cfp1^{d/d}* uterine cells. Thus, it is experimentally undoable to precipitate DNA fragments that CFP1 binds using CFP1 antibodies in *Cfp1^{d/d}* uterine cells. Alternatively, to meet the request of reviewer 2, we evaluated the specificity of CFP1 antibodies used in CFP1 ChIP-Seq even though it is an indirect way. We performed immunoprecipitation followed by Western blotting with CFP1 antibodies and provided the results in Supplementary Figure 10 in the revised manuscript. We clearly show that the CFP1 antibody binds explicitly to CFP1 protein in *Cfp1^{f/f}* but not in *Cfp1^{d/d}* uterus in the Supplementary Figure 10 of the revised manuscript (lines 483-495, lines 942-945). This result suggests that our CFP1 ChIP-Seq with only *Cfp1^{f/f}* uterine samples is informative, although it does not include *Cfp1^{d/d}* data.

We also reviewed literature with similar experimental settings, including CFP1 ChIP-Seq for *Cfp1^{d/d}* cells in other cell types, such as immune cells. They all performed CFP1 ChIP-Seq for *Cfp1^{f/f}* cells but not for *Cfp1^{d/d}* cells (Nat Commun 2016 May 23;7:11687. Doi:10.1038/ncomms11687; Sci Adv. 2019 Oct 9;5(10):eaax1608. Doi: 10.1126/sciadv.aax1608; eLife 2017 Apr 10. Doi:10.7554/eLife.24570; Proc Natl Acad Sci USA 2017 May 9;114(19):E3796-E3805. Doi:10.1073/pnas.1700909114; Sci Adv 2020 Jun 24;6(26):eaaz4764. Doi:10.1126/sciadv.aaz4764). These references clearly support why we could not perform CFP1 ChIP-Seq with *Cfp1^{d/d}* uterine samples.

3. Figure 2I; Add RT-PCR to the graph. Otherwise, the title atop is confusing.

→ **Author response:** We are thankful for the detailed review of our manuscript. We agree with the reviewer's comment. Therefore, we included the image of RT-PCR for the genes as well as graphs of real-time RT-PCR results in Figure 2I. In addition, we changed the position of the title of GO (Regulation of smoothed signaling pathway) from the top to the bottom of the graphs in Figure 2I in the revised manuscript (lines 155-156, line 821).

4. From human CFP1 data, we know that CFP1 occupies active CGI TSSs, but it is not restricted to them. Rather, it also occupies non-CGI TSSs and enhancers of actively transcribed genes. The authors have shown a significant portion of DEGs with decreased expression levels, but show increased H3K4me3 levels in *Cfp1d/d* mouse on Day 4, suggesting an H3K4me3-independent manner for CFP1 gene expression. They also show that the “number of significantly downregulated (3329) and upregulated (2829) genes in *Cfp1d/d* mice was comparable (54% vs. 46%) in DEGs from mRNA-seq. However, further analysis of gene ontology (GO) terms showed that 130 most GO terms (126/132, 95.5%) with false discovery rate (FDR) of <0.25 were reduced in the uteri of *Cfp1d/d* mice on Day 4.”

My understanding from the authors description here, is that even though they have seen comparable up- and down-regulation in their mRNA result, they are assuming more down-regulation than up-regulation based on their subsequent GO analysis. This interpretation may or may not be accurate. The GO algorithm, among other things, measures “representation” of terms, either over- or under-representation. This could also depend on the frequency in either an input background or sample frequencies. And may or may not reflect the underlying biology. Therefore, one could assume that not very frequent, but still significant, upregulated genes may not be “enriched” in the GO output. This would also include unresolved gene names. GO does provide these lists with the table of outputs. Note that the lack of assignment of a gene/ groups of genes by GO, in this case the upregulated genes, does not invalidate the mRNA expression data. Or their prevalence. The down-regulating

genes have shown a strong enrichment in specific functions/pathways, and it is of great interest; however, up-regulated genes, may be scattered across the genome, as I mentioned above, not at CGI TTSs, each with a potentially significant biological role, but without particular sets of genes to indicated specific functions/pathways/terms. Therefore, GO is not a downstream validating analysis to the mRNA expression data and the authors cannot assume a more prominent down-regulating role on the basis of their GO analysis. They must edit the wording in the text to reflect this.

→ **Author response:** We deeply appreciate your careful review and thoughtful explanation of this issue. We agree with reviewer 2 on every aspect that the reviewer commented on in a step-by-step manner. We understand why reviewer 2 was concerned that our wording may give a wrong impression on this issue. In the submitted manuscript, we showed that the numbers of up- and down-regulated genes were comparable in our mRNA-Seq. Most gene sets at statistical significance were down-regulated in GO analysis. This may mislead that GO analysis is associated with DEG analysis. Like reviewer 2 mentioned above, we also think that these two analyses are independent of each other. According to the comment of reviewer 2, we carefully looked through our text and rephrased the words to exclude any possibility that our GO analyses are associated with DEG analysis in the revised manuscript (line 134, lines 141-142).

5. Line 151: “Interestingly, the upstream and downstream genes of 149 IHH-dependent SSP were included in the list; *Ihh*, *Gli3*, and *Gata2* as H3K4me3-dependent and *Ptch1*, *Sox17*, and *Nr2f2* as H3K4me3-independent target genes (Fig. 3B and Supplementary Table 1). The expression of these genes was significantly reduced in *Cfp1d/d* mice (Figs. 2I and 3C)”. Unlike RT-PCRs in Fig 2I, in figure 3C, the relative expression change is statistically significantly changed, but not significantly. Meaning these relative expressions changes are not that significant/large, even if this reaches statistical significance. This wording must be corrected in the text for accuracy.

→ **Author response:** We deeply appreciate the detailed comment of reviewer 2 on the wording of the text in the submitted manuscript. According to the comment of reviewer 2, we carefully reviewed our text and rephrased the words not to exaggerate the meaning of our data, especially the extent of the difference between *Cfp1^{fl/fl}* and *Cfp1^{d/d}* uterine cells in the revised manuscript (lines 156-157); “The expression of these genes was reduced with statistical significance in *Cfp1^{d/d}* mice (Figs. 2I and 3C)”.

6. Also as in No 5, in text re Supp Fig 8B (line 204), this should be corrected to statistically significantly different. As expression levels are not that substantially different, even though reaching statistical significance.

→ **Author response:** As suggested by the reviewer above, we carefully looked at the text and rephrased the words accordingly in the revised manuscript (lines 211-213); “However, there was a statistically significant reduction in their expression levels in patients with severe endometriosis (Supplementary Fig. 9B)”.

7. I find the data on H3K4me3 on *Gata2*, *Ihh*, and *Sox17*, in *Cfp1d/d* mice and the subsequent validation of promoter enrichment of SETD1 and H3K4me3 of great interest. However, the enrichment at *sox17* is quite low in both mice models, and while it is likely that CFP1-SETD1 complex increases promoter H3K4me3 at *Ihh* and *Gata2*, I am not sure we can arrive at the conclusion, based on these data that CFP1 is the sole agent for *Sox17* expression. Please re-word this section for accuracy.

→ **Author response:** We are thankful for the critical comment of reviewer 2. Reviewer 2 precisely pinpointed that *Sox17* expression is H3K4me3-independent while it is regulated by the CFP1-SETD1 complex as we mentioned in the submitted manuscript. While reviewer 2 was concerned that we insist that CFP1 is the sole agent for *Sox17* expression, we did not imply that CFP1 is the sole agent for *Sox17* expression in the submitted manuscript. However, we admit that the sentence may make reviewer 2 misunderstand what we would like to explain with the results in Figure 3. In the submitted manuscript, we had the sentence; “In summary, CFP1 works with SETD1 to increase H3K4me3 in *Gata2* and *Ihh* promoters but acts alone for *Sox17* expression in the uterus. We think the two words “acts alone” may give a different meaning that the CFP1-SETD1 complex is the only one to control *Sox17* expression in the uterus, as reviewer 2 is concerned. Thus, we rephrased the sentence to avoid misunderstanding this issue (line 164); “In summary, CFP1 works with SETD1 to increase H3K4me3 in *Gata2* and *Ihh* promoters but not for *Sox17* promoter for their expression in the uterus” in the revised manuscript.

REVIEWERS' COMMENTS

Reviewer #1 (Remarks to the Author):

The authors have adequately addressed my concerns and the paper is improved accordingly.

Reviewer #2 (Remarks to the Author):

I find that the authors have adequately addressed my concerns.

A point-by-point response to the reviewer's comments

We deeply appreciate critical comments from the reviewers. All the comments from the reviewers are critical to strengthening the excellence of our manuscript. To meet all the requests from the reviewers, we have tried our best for performing suggested experiments and revising the manuscript. Newly added and revised words were colored yellow in the revised manuscript. We very much appreciate the opportunity to submit this revised manuscript and hope that, with these changes and clarifications, this work might be suitable for publication in Nature Communications.

Sincerely,

Haengseok Song, Ph.D.

Professor

Department of Biomedical Science

CHA University, Seongnam, Korea

Tel:82-031-881-7150

Fax:82-031-881-7249

Email: hssong@cha.ac.kr

Reviewer #1 (Remarks to the Author):

This study investigates a role of CFP1 on progesterone (P4) resistance using *Cfp1*^{flox/flox};*Pgr*-Cre (*Cfp1*^{d/d}) mice. *Cfp1*^{d/d} mice reveal several reproductive phenotypes in ovary, uterus and endometriosis model. The mutant mice are infertile due to implantation failure and non-receptive endometrium. Transcriptomic and ChIP-seq analysis identify P4 responsive genes including *Gata2*, *Sox17*, and *Ihh* as direct CFP1 targets. Interestingly, some phenotypes of mutant mice are rescued by a smoothed agonist. Furthermore, they confirm that CFP1 is significantly downregulated in women with endometriosis compared to control and expression levels between CFP1 and these P4 targets are positively related regardless of PGR levels. Based on strong results, they conclude that CFP1 is a critical epigenetic regulator for P4 signaling on uterine functions. The study is carefully designed and meticulously executed. The manuscript is well written in a clear manner and results support the conclusion. The findings are interesting and provide some insight into the regulation of progesterone signaling during pregnancy. Please clarify the following points for revision. However, there are some points to be addressed.

1. This study shows that CFP1 is a critical epigenetic regulator for P4 response using GATA2, SOX17, and IHH genes. There are a lot of P4 responsive genes. Have you compared your transcriptomic data to other P4-related transcriptomic data sets? It will give us more confident conclusion.

→ **Author response:** We appreciate a valuable comment from reviewer 1. As suggested by reviewer 1, we compared our datasets with other P₄-related transcriptomic datasets in different physiologic/experimental settings (GSE118264, GSE40661, and GSE178541) and provided comparison results as Supplementary Figure 7 in the revised manuscript. As shown in Supplementary Figure 7 in the revised manuscript, significant numbers of differentially expressed genes (DEGs) in our datasets (GSE219104) with *Cfp1*^{fl/fl} and *Cfp1*^{d/d} uterine samples are overlapped with those of DEGs in other P₄-related datasets, such as *Sox17* wildtype versus knockout uterus on Day 4 (GSE118264), *Gata2* wildtype versus knockout mice treated with P₄ (GSE40661), and *PgrA* and *PgrB* overexpression mice treated with oil or P₄ (GSE178541). We added a sentence to explain the results of comparative analyses in Supplementary Figure 7 in the revised manuscript (lines 175-178, lines 554-555, lines 923-928); "When our mRNA-seq datasets were compared with other P₄-related transcriptomic data (GSE118264, GSE40661, and GSE178541), comparative analyses showed that significant numbers of DEGs in *Cfp1*^{d/d} uterus overlap with those of DEGs in other P₄-related datasets (Supplementary Fig. 7)".

2. Strong CFP1 expressions are detected in ovary. Do *Cfp1* d/d mice have abnormal ovarian functions including ovulation, fertilization, and ovarian hormone production?

→ **Author response:** We agree with reviewer 1 on this issue. Since CFP1 is also expressed in the ovary, as mentioned by the reviewer, it is also important to examine the potential function of CFP1 in the ovary. Thus, we performed a series of experiments to examine whether the loss of CFP1 causes any aberration in the ovary as well as in the uterus. The revised manuscript includes all the data the reviewer mentioned, such as ovarian hormone levels, ovulation, fertilization, and in vitro development of embryos from *Cfp1*^{d/d} mice (Supplementary Figure 3D-F). We added sentences to demonstrate what we have in this Figure (lines 88-91, lines 103-104, lines 363-368, lines 370-376, lines 890-891, lines 897-903); "Furthermore, serum levels of E₂ and P₄ on Day 4 in *Cfp1*^{d/d} mice were comparable to those of *Cfp1*^{fl/fl} mice. We also found that *Cfp1*^{d/d} mice ovulate similar numbers of oocytes that can fertilize normally, and the fertilized embryos develop to the

blastocyst stage without any aberrations *in vitro* (Supplementary Fig. 3D-F)” in the revised manuscript. Therefore, we conclude that CFP1 is expressed in the ovary and the uterus, but the loss of CFP1 mainly disturbs uterine P₄ response, but not ovary function in mice.

3. Supplementary Fig. 2C: There are several CFP1 expressed cells in the uterus. Especially, uterine epithelial cells show high CFP1 expression in *Cfp1* d/d mice. Is there inefficient ablation of *Cfp1* in the mutant mice?

→ **Author response:** We are thankful for the thorough comments of reviewer 1 on the image in Supplementary Figure 2C. We agree with the reviewer’s comment on that image. The image may give a wrong impression that CFP1 may be inefficiently ablated by PR-Cre in *Cfp1*^{d/d} mice. Thus, to ensure that CFP1 is efficiently deleted in the uterus, we performed CFP1 immunostaining again in the *Cfp1*^{d/d} uterus during the revision period. The revised manuscript contains a new image of CFP1 immunostaining in Supplementary Figure 2C to take away the concern that CFP1 may not be efficiently deleted.

4. Fig. 3H: Are the recovery rates of P₄ target genes by SAG compatible to wild type mice? Add wild type or f/f control mice treated with PBS and SAG to the rescue experiment.

→ **Author response:** We deeply appreciate the critical comment of reviewer 1 on this experiment. We also think it is scientifically more relevant to include groups of *Cfp1*^{f/f} mice treated with PBS and SAG as controls for the experiment to rescue the phenotypes of *Cfp1*^{d/d} mice with SAG. According to the reviewer’s comment, we performed a new experiment to include two control groups (PBS and SAG treatment) in *Cfp1*^{f/f} mice during the revision period (n=8 per each group) and have a new Figure 3H in the revised manuscript (lines 167-170, lines 174-175, lines 428-430).

In the submitted manuscript, although we found that SAG significantly increased expression levels of these P₄ target genes, such as *Gli1*, *Gli2*, *Nr2f2*, and *Hand2*, in *Cfp1*^{d/d} uterus, we were not able to quantitatively evaluate how much SAG rescues the reduced expression levels of these genes. In the revised Figure 3H, we are now able to clearly mention that SAG supplementation can efficiently restore expression levels of the P₄ target genes in *Cfp1*^{d/d} uterus as comparable as that of *Cfp1*^{f/f} uterus with or without SAG. In addition, we can provide that SAG treatment did not have any additional effects on the P₄-dependent expression of these P₄ target genes in *Cfp1*^{f/f} uterus.

5. Line 5: Rewrite the sentence. There is strong relationship between infertility and endometriosis. However, we have not known the mechanisms including progesterone resistance.

→ **Author response:** We agree with the reviewer on this issue. Progesterone (P₄) resistance is a cause of various endometrial disorders, such as endometriosis, infertility, inflammatory disorders, and cancer. Endometriosis affects 10%–15% of women of reproductive age and is one of the major causes of female infertility. Thus, the sentence that reviewer 1 pointed out may not be a good one to describe the relationship between progesterone resistance, endometriosis, and infertility. According to the comment, we revised the sentence to make the relationship between them more straightforward (lines 5-6); “P₄ resistance is a major cause of the pathogenesis of endometrial disorders like endometriosis, often leading to infertility” in the revised manuscript.

6. Line 19; Provide specific conditions for physiologic and pathophysiologic conditions.

→ **Author response:** According to reviewer 1's comment, we mentioned specific events for physiologic and pathophysiologic conditions in the abstract. Therefore, the revised sentence is "CFP1 is a key epigenetic factor that intervenes in the P₄-epigenome-transcriptome networks for uterine receptivity for embryo implantation and the pathogenesis of endometriosis" in the revised manuscript (lines 18-19).

Reviewer #2 (Remarks to the Author):

I congratulate the authors for this manuscript. I believe their findings will help advance our understanding of both the uterine epigenetic machinery and in the pathophysiology of endometriosis. They have shown in the *Cfp1*^{flox/flox};Pgr-Cre mouse model that while *cfp1* is absent in most cells of mouse adult FRT, this does not affect the morphology of cells yet normal decidualization, blastocyst implantation, implantation sites and live pups were aberrant, indicating a deficient uterine microenvironment. They then provide further evidence on the P4-CFP1 interplay both dependent and independent of H3K4me3, and via IHH-dependent pathway for this aberrancy. They examined both stromal and epithelia cells and by using a Smo agonist. For potential role of CFP1 in endometriosis they use a mouse model to assess P4-response via CFP1 in ectopic lesion growth. Evaluating GATA2, SOX17, and IHH mRNA levels (and CFP1 and PGR) in a human mild, severe and non-disease dataset they suggest a potential role of CFP1 in suggested P4-resistance in endometriosis.

1. I do agree that their data show an important role for CFP1 in aberrant epigenetic landscape in patients with endometriosis but, both their mouse model and the human subset they have evaluated are models of existing lesions and does not provide evidence for a “leading” role for CFP1 (as they have suggested in the end of the Results section). This should be re-worded. Why and how some women develop endometriosis, from immune evasion, to attachment in the peritoneum/on organs, to sustained and expanded growth, the potential role of CFP1 in each of these are yet to be determined, and we cannot yet assume a leading role for CFP1 in endometriosis. Besides the above point, the following few points are needed to be addressed to ensure optimum accuracy of the presented manuscript:

→ **Author response:** We absolutely agree with reviewer 2 on this issue. However, we do not think that we “DO” provide evidence that CFP1 has a leading role in the pathogenesis of endometriosis. As mentioned by reviewer 2, endometriosis is a multifactorial disease that is driven by complex genetic, physiologic, and environmental interactions. In the submitted manuscript, we provide a clue to understanding the underlying epigenetic mechanism(s) that could cause progesterone resistance that may promote the development of endometriosis in mice and humans. Reviewer 2 got the impression that we imply that CFP1 has “a leading role” in the pathogenesis of endometriosis in humans because of the last subtitle in the Result section, “Downregulation of the epigenetic factor CFP1 may cause P₄ resistance, leading to endometriosis in humans”. We agree with reviewer 2 that the subtitle may overstate what we found in the submitted manuscript. We did not mean to state that aberrant down-regulation of CFP1 expression in human endometrium could be a leading cause of endometriosis. Thus, we tone-downed the last subtitle to “Downregulation of the epigenetic factor *CFP1* may be associated with endometriosis in humans” accordingly in the revised manuscript (lines 200-201).

2. In Figure 2A the authors have CFP1 ChIP-Seq for *f/f* mice, but also need to include for *d/d* mice (no IP would be expected, but it is necessary to show for comparison). This is important because the aim of this experiment was to show that H3K4me3 was reduced (in a CFP1-dependent manner) in TSS and CGI in *Cfp1*^{d/d} mouse uterus, due to lack of CFP1.

→ **Author response:** We absolutely agree with reviewer 2 on this issue. We also had the same concern that reviewer 2 mentioned from the very beginning. Initially, we were going to send not only *Cfp1*^{f/f} but also *Cfp1*^{d/d} uterine samples for CFP1 ChIP-Seq. However, the company that performed the sequencing with

our samples suggested going with only *Cfp1^{f/f}* uterine samples. The reason is that the library for ChIP-Seq can be constructed with the DNA segments precipitated by CFP1 antibodies in *Cfp1^{d/d}* uterine cells but not in *Cfp1^{d/d}* uterine cells since CFP1 protein barely exists in *Cfp1^{d/d}* uterine cells. Thus, it is experimentally undoable to precipitate DNA fragments that CFP1 binds using CFP1 antibodies in *Cfp1^{d/d}* uterine cells. Alternatively, to meet the request of reviewer 2, we evaluated the specificity of CFP1 antibodies used in CFP1 ChIP-Seq even though it is an indirect way. We performed immunoprecipitation followed by Western blotting with CFP1 antibodies and provided the results in Supplementary Figure 10 in the revised manuscript. We clearly show that the CFP1 antibody binds explicitly to CFP1 protein in *Cfp1^{f/f}* but not in *Cfp1^{d/d}* uterus in the Supplementary Figure 10 of the revised manuscript (lines 483-495, lines 942-945). This result suggests that our CFP1 ChIP-Seq with only *Cfp1^{f/f}* uterine samples is informative, although it does not include *Cfp1^{d/d}* data.

We also reviewed literature with similar experimental settings, including CFP1 ChIP-Seq for *Cfp1^{d/d}* cells in other cell types, such as immune cells. They all performed CFP1 ChIP-Seq for *Cfp1^{f/f}* cells but not for *Cfp1^{d/d}* cells (Nat Commun 2016 May 23;7:11687. Doi:10.1038/ncomms11687; Sci Adv. 2019 Oct 9;5(10):eaax1608. Doi: 10.1123/sciadv.aax1608; eLife 2017 Apr 10. Doi:10.7554/eLife.24570; Proc Natl Acad Sci USA 2017 May 9;114(19):E3796-E3805. Doi:10.1073/pnas.1700909114; Sci Adv 2020 Jun 24;6(26):eaaz4764. Doi:10.1126/sciadv.aaz4764). These references clearly support why we could not perform CFP1 ChIP-Seq with *Cfp1^{d/d}* uterine samples.

3. Figure 2I; Add RT-PCR to the graph. Otherwise, the title atop is confusing.

→ **Author response:** We are thankful for the detailed review of our manuscript. We agree with the reviewer's comment. Therefore, we included the image of RT-PCR for the genes as well as graphs of real-time RT-PCR results in Figure 2I. In addition, we changed the position of the title of GO (Regulation of smoothed signaling pathway) from the top to the bottom of the graphs in Figure 2I in the revised manuscript (lines 155-156, line 821).

4. From human CFP1 data, we know that CFP1 occupies active CGI TSSs, but it is not restricted to them. Rather, it also occupies non-CGI TSSs and enhancers of actively transcribed genes. The authors have shown a significant portion of DEGs with decreased expression levels, but show increased H3K4me3 levels in *Cfp1d/d* mouse on Day 4, suggesting an H3K4me3-independent manner for CFP1 gene expression. They also show that the “number of significantly downregulated (3329) and upregulated (2829) genes in *Cfp1d/d* mice was comparable (54% vs. 46%) in DEGs from mRNA-seq. However, further analysis of gene ontology (GO) terms showed that 130 most GO terms (126/132, 95.5%) with false discovery rate (FDR) of <0.25 were reduced in the uteri of *Cfp1d/d* mice on Day 4.”

My understanding from the authors description here, is that even though they have seen comparable up- and down-regulation in their mRNA result, they are assuming more down-regulation than up-regulation based on their subsequent GO analysis. This interpretation may or may not be accurate. The GO algorithm, among other things, measures “representation” of terms, either over- or under-representation. This could also depend on the frequency in either an input background or sample frequencies. And may or may not reflect the underlying biology. Therefore, one could assume that not very frequent, but still significant, upregulated genes may not be “enriched” in the GO output. This would also include unresolved gene names. GO does provide these lists with the table of outputs. Note that the lack of assignment of a gene/ groups of genes by GO, in this case the

upregulated genes, does not invalidate the mRNA expression data. Or their prevalence. The down-regulating genes have shown a strong enrichment in specific functions/pathways, and it is of great interest; however, up-regulated genes, may be scattered across the genome, as I mentioned above, not at CGI TTSs, each with a potentially significant biological role, but without particular sets of genes to indicated specific functions/pathways/terms. Therefore, GO is not a downstream validating analysis to the mRNA expression data and the authors cannot assume a more prominent down-regulating role on the basis of their GO analysis. They must edit the wording in the text to reflect this.

→ **Author response:** We deeply appreciate your careful review and thoughtful explanation of this issue. We agree with reviewer 2 on every aspect that the reviewer commented on in a step-by-step manner. We understand why reviewer 2 was concerned that our wording may give a wrong impression on this issue. In the submitted manuscript, we showed that the numbers of up- and down-regulated genes were comparable in our mRNA-Seq. Most gene sets at statistical significance were down-regulated in GO analysis. This may mislead that GO analysis is associated with DEG analysis. Like reviewer 2 mentioned above, we also think that these two analyses are independent of each other. According to the comment of reviewer 2, we carefully looked through our text and rephrased the words to exclude any possibility that our GO analyses are associated with DEG analysis in the revised manuscript (line 134, lines 141-142).

5. Line 151: “Interestingly, the upstream and downstream genes of 149 IHH-dependent SSP were included in the list; *Ihh*, *Gli3*, and *Gata2* as H3K4me3-dependent and *Ptch1*, *Sox17*, and *Nr2f2* as H3K4me3-independent target genes (Fig. 3B and Supplementary Table 1). The expression of these genes was significantly reduced in *Cfp1d/d* mice (Figs. 2I and 3C)”. Unlike RT-PCRs in Fig 2I, in figure 3C, the relative expression change is statistically significantly changed, but not significantly. Meaning these relative expressions changes are not that significant/large, even if this reaches statistical significance. This wording must be corrected in the text for accuracy.

→ **Author response:** We deeply appreciate the detailed comment of reviewer 2 on the wording of the text in the submitted manuscript. According to the comment of reviewer 2, we carefully reviewed our text and rephrased the words not to exaggerate the meaning of our data, especially the extent of the difference between *Cfp1^{fl/fl}* and *Cfp1^{d/d}* uterine cells in the revised manuscript (lines 156-157); “The expression of these genes was reduced with statistical significance in *Cfp1^{d/d}* mice (Figs. 2I and 3C)”.

6. Also as in No 5, in text re Supp Fig 8B (line 204), this should be corrected to statistically significantly different. As expression levels are not that substantially different, even though reaching statistical significance.

→ **Author response:** As suggested by the reviewer above, we carefully looked at the text and rephrased the words accordingly in the revised manuscript (lines 211-213); “However, there was a statistically significant reduction in their expression levels in patients with severe endometriosis (Supplementary Fig. 9B)”.

7. I find the data on H3K4me3 on *Gata2*, *Ihh*, and *Sox17*, in *Cfp1d/d* mice and the subsequent validation of promoter enrichment of SETD1 and H3K4me3 of great interest. However, the enrichment at *sox17* is quite low in both mice models, and while it is likely that CFP1-SETD1 complex increases promoter H3K4me3 at *Ihh* and *Gata2*, I am not sure we can arrive at the conclusion, based on these data that CFP1 is the sole agent for

Sox17 expression. Please re-word this section for accuracy.

→ **Author response:** We are thankful for the critical comment of reviewer 2. Reviewer 2 precisely pinpointed that *Sox17* expression is H3K4me3-independent while it is regulated by the CFP1-SETD1 complex as we mentioned in the submitted manuscript. While reviewer 2 was concerned that we insist that CFP1 is the sole agent for *Sox17* expression, we did not imply that CFP1 is the sole agent for *Sox17* expression in the submitted manuscript. However, we admit that the sentence may make reviewer 2 misunderstand what we would like to explain with the results in Figure 3. In the submitted manuscript, we had the sentence; “In summary, CFP1 works with SETD1 to increase H3K4me3 in *Gata2* and *Ihh* promoters but acts alone for *Sox17* expression in the uterus. We think the two words “ACTS ALONE” may give a different meaning that the CFP1-SETD1 complex is the only one to control *Sox17* expression in the uterus, as reviewer 2 is concerned. Thus, we rephrased the sentence to avoid misunderstanding this issue (line 164); “In summary, CFP1 works with SETD1 to increase H3K4me3 in *Gata2* and *Ihh* promoters but not for *Sox17* promoter for their expression in the uterus” in the revised manuscript.